# An experimentally supported model of the *Bacillus subtilis* global transcriptional regulatory network

Mario L Arrieta-Ortiz[1,†], Christoph Hafemeister[1,†], Ashley Rose Bate[1], Timothy Chu[1], Alex Greenfield[1], Bentley Shuster[1], Samantha N Barry[1], Matthew Gallitto[1], Brian Liu[1], Thadeous Kacmarczyk[1], Francis Santoriello[1], Jie Chen[1], Christopher DA Rodrigues[2], Tsutomu Sato[3], David Z Rudner[2], Adam Driks[4], Richard Bonneau[1,5,6,*] & Patrick Eichenberger[1,**]

## Abstract

Organisms from all domains of life use gene regulation networks to control cell growth, identity, function, and responses to environmental challenges. Although accurate global regulatory models would provide critical evolutionary and functional insights, they remain incomplete, even for the best studied organisms. Efforts to build comprehensive networks are confounded by challenges including network scale, degree of connectivity, complexity of organism–environment interactions, and difficulty of estimating the activity of regulatory factors. Taking advantage of the large number of known regulatory interactions in *Bacillus subtilis* and two transcriptomics datasets (including one with 38 separate experiments collected specifically for this study), we use a new combination of network component analysis and model selection to simultaneously estimate transcription factor activities and learn a substantially expanded transcriptional regulatory network for this bacterium. In total, we predict 2,258 novel regulatory interactions and recall 74% of the previously known interactions. We obtained experimental support for 391 (out of 635 evaluated) novel regulatory edges (62% accuracy), thus significantly increasing our understanding of various cell processes, such as spore formation.

**Keywords** *Bacillus subtilis*; network inference; sporulation; transcriptional networks

**Subject Categories** Genome-Scale & Integrative Biology; Transcription; Network Biology

**Mol Syst Biol. (2015) 11: 839**

## Introduction

As cells navigate their environment, divide and differentiate, they rely on gene regulation networks. Building accurate and comprehensive models of the interactions between regulators and their target genes is essential to our understanding of basic biology and evolution in all living systems (Marbach *et al*, 2012). Analysis of gene regulatory networks can reveal how diverse cell processes are balanced in a single organism and facilitate genome annotation by uncovering hidden functions of co-regulated genes. Global gene regulation networks also provide information on a system's robustness and evolvability, thus influencing bioengineering strategies. In spite of the importance of having complete models, the fraction of known regulatory interactions is quite small for most species, with well-studied organisms having at best half of their genes paired with a regulator (Salgado *et al*, 2013). The lack of any completely characterized network stems from the fact that experimental assays are limited to measuring the consequences of gene regulation (e.g. changes in RNA or protein levels) or assessing the binding of regulators to promoters or mRNAs (Hughes & de Boer, 2013). The rate at which new regulatory interactions are identified was, however, significantly accelerated with the advent of genomic technologies (Salgado *et al*, 2013).

Here, we focus on *Bacillus subtilis,* a model organism for the human pathogens *Bacillus anthracis, Clostridium difficile, Listeria monocytogenes,* and *Staphylococcus aureus. B. subtilis* was the first Gram-positive bacterium to have its genome sequenced (Kunst *et al*, 1997; Barbe *et al*, 2009) and is a major model system for competence, biofilm, and spore formation (Dubnau & Mirouze, 2013; Vlamakis *et al*, 2013; Cairns *et al*, 2014; Tan & Ramamurthi, 2014). Sporulation, in particular, is among the best-understood developmental processes in biology. The main regulators of gene expression during sporulation are σ factors, subunits of the RNA polymerase

1  Center for Genomics and Systems Biology, Department of Biology, New York University, New York, NY, USA
2  Department of Microbiology and Immunobiology, Harvard Medical School, Boston, MA, USA
3  Department of Frontier Bioscience, Hosei University, Koganei, Tokyo, Japan
4  Department of Microbiology and Immunology, Stritch School of Medicine, Loyola University Chicago, Maywood, IL, USA
5  Courant Institute of Mathematical Science, Computer Science Department, New York, NY, USA
6  Simons Foundation, Simons Center for Data Analysis, New York, NY, USA
   *Corresponding author. Tel: +1 212 992 9516; E-mail: rbonneau@simonsfoundation.org
   **Corresponding author. Tel: +1 212 998 8247; E-mail: pe19@nyu.edu
   †These authors contributed equally to this work

conferring DNA binding specificity to the holoenzyme (Stragier & Losick, 1990; Feklístov *et al*, 2014). Asymmetric division of the sporulating cell results in two cell types, a forespore that will mature into a spore and a larger mother cell. Forespore maturation depends on the mother cell for metabolic activity and synthesis of more than 70 spore coat proteins (McKenney *et al*, 2013). Many of the original transcriptomic studies in *B. subtilis* focused on gene expression during sporulation and other stress responses (Fawcett *et al*, 2000; Cao *et al*, 2002; Price *et al*, 2002; De Hoon *et al*, 2010). More recently, an expanded transcriptomic dataset was collected (Nicolas *et al*, 2012). An overarching goal in *B. subtilis* systems biology is to integrate these datasets with quantitative proteomics (Soufi *et al*, 2010) and analyses of metabolic fluxes (Chubukov *et al*, 2013) to obtain a comprehensive model of gene regulation, similar to what has been accomplished in a multi-omics investigation of the glucose to malate metabolic shift (Buescher *et al*, 2012).

Previous regulatory network inference efforts can be divided into three main categories: (i) curation of literature and transcription factor (TF) binding sites (Freyre-González *et al*, 2013; Leyn *et al*, 2013); (ii) genetic perturbations to learn directed edges (Madar *et al*, 2010; Ciofani *et al*, 2012); and (iii) modeling regulation as a dynamic process using time series data (Bonneau *et al*, 2006). To reduce the complexity of the problem, regulatory networks have often been determined for TFs controlling gene clusters, instead of individual genes (Fadda *et al*, 2009; Lemmens *et al*, 2009; Waltman *et al*, 2010; Brooks *et al*, 2014; Peterson *et al*, 2015; Reiss *et al*, 2015), and metabolic pathways were integrated with regulatory networks (Oh *et al*, 2007; Goelzer *et al*, 2008; Labhsetwar *et al*, 2013; O'Brien *et al*, 2013; Carrera *et al*, 2014). As a whole, prior network inference studies improved prediction of the effects of genetic perturbations or the accumulation rates of metabolites under different growth conditions (Imam *et al*, 2015; Kim *et al*, 2015). In spite of early successes (Faith *et al*, 2007), most studies remained limited in a number of ways and often relied on heterogeneous datasets (e.g. using various microarray platforms and strain backgrounds). For instance, predicted networks for *E. coli* and *B. subtilis* were less complex than the prior known networks (i.e. these studies did not expand the known networks substantially, but instead highlighted a small focused set of new and known edges). Furthermore, in most cases, the accuracy of novel predictions was not systematically assessed in follow-up experiments.

Network inference is a difficult problem because of (i) biological complexity (the activity of a transcription factor (TF) is not linearly related to its abundance); (ii) non-identifiability (biological networks are robust and thus many potential models will explain any given dataset equally well); and (iii) systematic error. Although complexity and measurement error constitute the two most often cited challenges, non-identifiability is perhaps a greater problem (Marbach *et al*, 2012). To address this issue, we described methods for learning regulatory networks that use prior knowledge on network structure to improve accurate identification of large networks (BBSR: Bayesian Best Subset Regression; Greenfield *et al*, 2013). Here, we use known interactions gathered from SubtiWiki (Michna *et al*, 2014) to both estimate TF activities (TFA) by employing network component analysis (NCA) (Liao *et al*, 2003) and constrain the model selection step of our method (BBSR-TFA).

Estimated TFA and NCA have been used in previous network inference efforts. For example, chromatin immuno-precipitation (ChIP) binding data were used to estimate *S. cerevisae* TFA followed by correlation for target identification (Gao *et al*, 2004). Similarly, interactions from RegulonDB (Salgado *et al*, 2013) were used to determine dynamics of activities of *E. coli* TFs during carbon source transition (Kao *et al*, 2004). Following these initial applications, numerous methods to estimate TFA and to learn the regulatory network have been proposed (Boulesteix & Strimmer, 2005; Galbraith *et al*, 2006; Sanguinetti *et al*, 2006; Gu *et al*, 2007; Fu *et al*, 2011; Noor *et al*, 2014). These methods have in common that they model gene expression to be the result of the connectivity strength between TF–gene pairs and TF activity, where the activity is a latent variable pooling the effects of post-transcriptional and post-translational modifications. More recent applications include the identification of key TFs and their targets in mice during rapamycin treatment (Tran *et al*, 2010), and a regulatory network important in floral development in *A. thaliana* (Misra & Sriram, 2013). To our knowledge, there is only one previous application of NCA to *B. subtilis* data (Buescher *et al*, 2012). In that work, known transcriptional regulation was taken from literature, the DBTBS (Sierro *et al*, 2008) and SubtiWiki databases, and CcpA ChIP-chip data, to learn the regulatory network perturbed during change of carbon substrate from glucose to malate and vice versa. However, as the main focus was on metabolism, none of the 1,488 predicted interactions were assessed in follow-up experiments. In this work, we apply a unified new combination of NCA and model selection to an experimental design expressly conceived to dynamically probe the principal cellular pathways of *B. subtilis*, and we identify 2,258 novel regulatory interactions of unprecedented accuracy.

## Results and Discussion

### A compendium of *B. subtilis* transcriptional profiling data

Our goal is to infer the transcriptional regulatory network (TRN) from two large transcriptomic datasets, while also incorporating previously validated TF–target gene interactions (Fig 1). These known regulatory interactions, compiled in SubtiWiki (Michna *et al*, 2014), represent a large body of prior work using several experimental methods to characterize functional regulatory links and are thus a powerful complement to the transcription compendium described below. We collected a global gene transcription compendium for strain PY79, a derivative of strain 168 frequently used for transcriptional profiling (Fawcett *et al*, 2000; Eichenberger *et al*, 2004; Wang *et al*, 2006), and obtained transcriptional profiles for 4,002 protein-coding genes from a total of 403 samples in 38 separate experimental designs (Table EV1 for strains used, Table EV2 for experimental conditions, GEO accession number GSE67023). A large fraction of the data was collected as time series, which improves our ability to infer directed edges (Bonneau *et al*, 2006). We investigated an entire life cycle from spore germination to sporulation (with samples collected at 30-min intervals), as well as stress responses, competence, and biofilm formation (complete results in Dataset EV1). We added a previously published dataset with 269 samples covering 104 conditions, using strain BSB1, another derivative of strain 168 (Nicolas *et al*, 2012). In order to incorporate informative priors on network structure, we retrieved from SubtiWiki a list of 3,040 experimentally

    

**Figure 1.   General workflow for inferring the *B. subtilis* transcription network.**
Two transcriptomic data compendia were used, one collected specifically for this study (strain PY79) and one previously published for strain BSB1 (Nicolas *et al*, 2012). Transcription factor activities (TFA) were estimated independently for each dataset using interactions in the gold standard (GS) extracted primarily from SubtiWiki (step 1). Datasets, estimated TFA, and priors on network structure (from the GS) were used as inputs for prediction of regulatory interactions (step 2). Next, output networks (one for each strain/dataset) were merged into a combined network (inferred TRN) (step 3) and prediction accuracy was evaluated in follow-up experiments.

validated regulatory interactions (Dataset EV2). Subsequently, we refer to this set of interactions as the "gold standard" (GS) or "prior network".

**Estimating transcription factor activities (TFA) increases the accuracy of network inference**

To learn the *B. subtilis* TRN, we used a new combination of our *Inferelator-BBSR* approach (Greenfield *et al*, 2013), with a method for estimating transcription factor activities (TFA). Previously, gene-specific regulators were discovered based on mRNA abundance correlation, that is, gene transcription profiles were modeled as linear combinations of one or a few TF transcription profiles. With our new dataset, we observed that transcription profile is not an optimal proxy for a TF's regulatory strength (Fig 2). As a consequence, we modified the procedure by introducing an initial step where TFA are estimated based on known regulatory interactions for each experimental condition. To do so, we used NCA (Liao *et al*, 2003) with a simplified model of transcriptional regulation compared to previous work (Kao *et al*, 2004; Buescher *et al*, 2012)

(details given in method section). Conceptually, this is similar to estimating TFA with a reporter gene, although here every known target of a TF is used as reporter and we explicitly model activation, repression, and genes under multi-TF control. These TFA were then used as predictors to learn the strength and sign of TF–gene interactions. Subsequently, predictions for each dataset were integrated into a combined network, where potential interactions were ranked by their confidence scores to provide networks that meet specific accuracy requirements (calibrated using known interactions).

*Motivation for estimating TFA*
In Fig 2A (top panel), we plot the partial correlation between transcription of each TF and known target gene along all conditions in the PY79 compendium. We observe that many TF–target pairs are only moderately correlated. This is expected, as a gene may be controlled by multiple regulators, while interactions of a TF with its targets are typically restricted to a subset of experimental conditions where the TF is expressed and active. We also note a high proportion of known negative interactions (repression) with positive

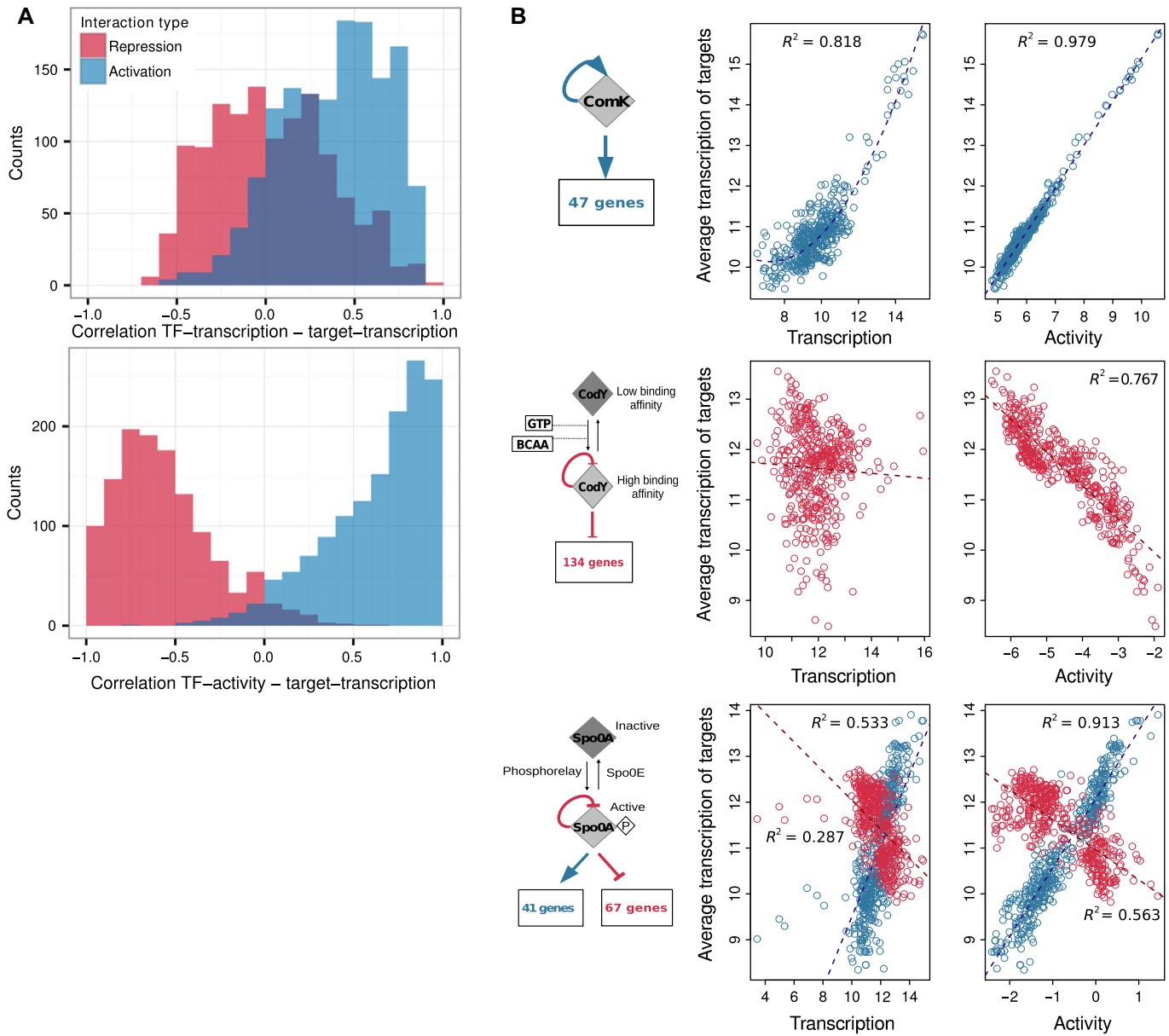

**Figure 2. Incorporating Transcription Factor Activities (TFA) in the network inference procedure.**

A Partial Pearson correlation between mRNA transcription levels (PY79 dataset) was computed for each TF–target gene pair in the GS (top histogram). Partial Pearson correlation was also computed between the estimated activity of a TF and the transcription of its targets (bottom histogram).

B The advantage of estimating TFA is illustrated for three regulators. Each point corresponds to the results of one microarray experiment, and TFA are estimated for each experimental condition. Top panel: A nonlinear correlation is observed between *comK* transcription and transcription of ComK targets, whereas a strong linear correlation is obtained between ComK activity and transcription of ComK targets. Middle panel: No correlation is observed between *codY* transcription and transcription of CodY targets. CodY activity is modulated by GTP and branched chain amino acids (BCAA). A negative correlation is observed between estimated CodY activity and transcription of CodY targets. Bottom panel: Spo0A activity is modulated by phosphorylation. A better correlation is observed between Spo0A activity and transcription of Spo0A targets than between *spo0A* transcription and transcription of Spo0A targets.

correlation scores. This can happen when repressors work as components of negative feedback mechanisms. For instance, when a repressor takes part in an incoherent feed forward loop (FFL), expression of the repressor will start at the same time as its targets and the negative effect on target expression will only be sensed after a delay necessary for the accumulation of the repressor (Mangan & Alon, 2003; Alon, 2007). By contrast, correlations between

estimated TFA and target gene transcription show fewer interactions with low correlation and better separation between activating and repressing interactions (Fig 2A bottom panel).

We examined the relationships between TF activity and target gene transcription in a group of 50 TFs with at least ten experimentally validated targets (Dataset EV3). Nonlinear relationships were detected for almost all regulators, including the master regulator of

competence ComK (Fig 2B top panel), which is subjected to a positive auto-regulatory loop and binds to its target promoters as dimer (Hamoen *et al*, 1998; Hamoen, 2003). The absence of a linear relationship between transcription of TFs and their targets was frequently noted for regulators that require co-factors, such as CodY (Sonenshein, 2007), a repressor whose activity is modulated by GTP and branched chain amino acids (Fig 2B middle panel). By contrast, for both ComK and CodY, a linear relationship is observed between TF activity and transcription of their targets. This linear relationship is a consequence of the way TFA are estimated (see Materials and Methods). Considering that the *Inferelator* is based on a linear model (see Materials and Methods), this linearization step is likely to improve the detection of additional regulatory interactions. This improvement would affect primarily TFs whose activity can be accurately estimated [i.e. those with > 10 known target genes, see below and Appendix Fig S1 (for the BSB1 data compendium) and Appendix Fig S2 (for the PY79 data compendium)]. Another major reason for discrepancies between TF transcription and target gene transcription is caused by post-translational modifications, such as the phosphorylation of response regulators in two-component systems (Salazar & Laub, 2015). A classic example is Spo0A, the master regulator of sporulation (Molle *et al*, 2003), which is activated by a phosphorelay (Fig 2B bottom panel). As Spo0A is both an activator and a repressor of transcription, we note that an efficient way to discriminate activated from repressed targets is to encode the sign of the interactions in the prior. Overall, estimation of TFA greatly increases the predictive power of the *Inferelator*.

### Performance of our network prediction framework

Given TFA estimates, we must still select the most probable regulatory network model from the transcription data. This very large-scale model selection step also leverages the dynamical information built into our experimental design. We use TFA as predictors (and rely on transcription profiles for TFs without known targets) to infer the global TRN with our framework for regulatory network model selection (*Inferelator*-BBSR). We analyzed the performance of our approach by (i) assessing the recovery of known regulatory edges; (ii) assessing the robustness to noise; (iii) evaluating the experimental support for the predictions; and (iv) comparing to other network inference methods. To assess the recovery of known interactions, we compute the Area Under the Precision Recall (AUPR) curve (Fig 3A) for this recovery task under both settings (recovery of known and learning of new interactions); AUPR has a value of 1 when all GS interactions rank top of the list and close to 0 for random predictions. In addition to the *Inferelator*, we also evaluated CLR (Faith *et al*, 2007) and Genie3 (Huynh-Thu *et al*, 2010), two state-of-the-art network inference methods. In this analysis, GS interactions were only used for TFA estimation and we did not incorporate prior information during the model selection step of the *Inferelator*. A random set of 50% of the GS interactions were used for TFA estimation, while the remaining GS interactions were used to calculate precision and recall. We note that (i) a higher score is obtained for the combined network than for networks derived from each dataset independently; (ii) scores for the predicted networks are significantly higher when TFA is used; and (iii) although priors were not used to influence model selection, the *Inferelator* has the highest AUPR among the compared methods.

To determine the stability of the estimated TFA, we examined the effect that changes in the set of GS interactions had on estimated TFA by randomly removing 20% of the GS interactions 128 times. The vast majority of TFA are stable [as indicated by the distributions of the pair-wise correlations of the activities; Appendix Fig S1 (for the BSB1 data compendium) and Appendix Fig S2 (for the PY79 data compendium)], and TFs with ten or more priors have more stable estimated activities than TFs with < 10 priors. This implies that excluding part of the GS network during TFA estimation does not have a significant effect on the activities of those TFs with dozens of targets. Next, to evaluate if the number of bootstraps affected the output of the inference approach, we compared the top 5,000 interactions for inferred networks using 2 up to 100 bootstraps in the BSB1 dataset, PY79 dataset, or both (combined) to the top 5,000 interactions using one less bootstrap (Appendix Fig S3). We observed that, for all networks, after 20 bootstraps, more than 4,890 (97.8%) are shared when another bootstrap is added. This finding suggests a rapid convergence of the error estimates computed by BBSR-TFA.

To assess robustness to noise (i.e. presence of incorrect or irrelevant edges in the GS), we used as priors 50% of the GS interactions (randomly selected) and added various amounts of random false interactions (Fig 3B). Performance on the remaining 50% of GS interactions demonstrates a very high error-tolerance and relative insensitivity to input parameters. Specifically, AUPR in the presence of a noisy prior is higher than the no-prior baseline even at a true:false prior ratio of 1:10, if weight in the model selection step (g-prior in the Materials and Methods section) is not too large (i.e. < 2). We conclude that using estimated TFA combined with constrained network model selection (BBSR-TFA) results in networks that are more consistent with prior knowledge and have increased accuracy. The fact that the combined network has the highest AUPR in Fig 3B indicates that many true interactions that would have been excluded otherwise are recovered from the combination of the PY79 and BSB1 independently predicted networks. This highlights the complementary nature of the two data compendia. In principle, our method can be applied to a variety of systems, as long as a set of priors is available.

To evaluate the experimental support for BBSR-TFA's predictions, we used transcriptional profiles we collected for the principal σ factors ($\sigma^B$, $\sigma^D$, $\sigma^E$, $\sigma^F$, $\sigma^G$, $\sigma^H$, $\sigma^K$, $\sigma^L$, $\sigma^M$, $\sigma^W$) and global regulators (AbrB, CodY, and Spo0A). In addition, we collected data for ComK (competence), SinR (biofilm formation), ScoC (transition to stationary phase), and PhoP (phosphate metabolism). For each selected TF, we measured the transcription rate of all genes in the wild-type (WT) and TF knockout (KO) strains. To have a clear separation between training and evaluation datasets, we predicted a network for each analyzed regulon using training data that excluded data relevant for the WT and KO strains' comparison for that TF. In total, we obtained 17 networks (Fig 4), one for each TF with KO data (we refer to these networks as evaluation networks). The reason for not excluding all KO data at once is that it would represent a 25% decrease in the PY79 dataset size.

For each TF and the corresponding set of KO conditions, we tested all genes for differential transcription (DT) using Bayesian *t*-tests. We considered all genes with *P*-values < 0.01 as DT (see Materials and Methods for details). Genes that were predicted as

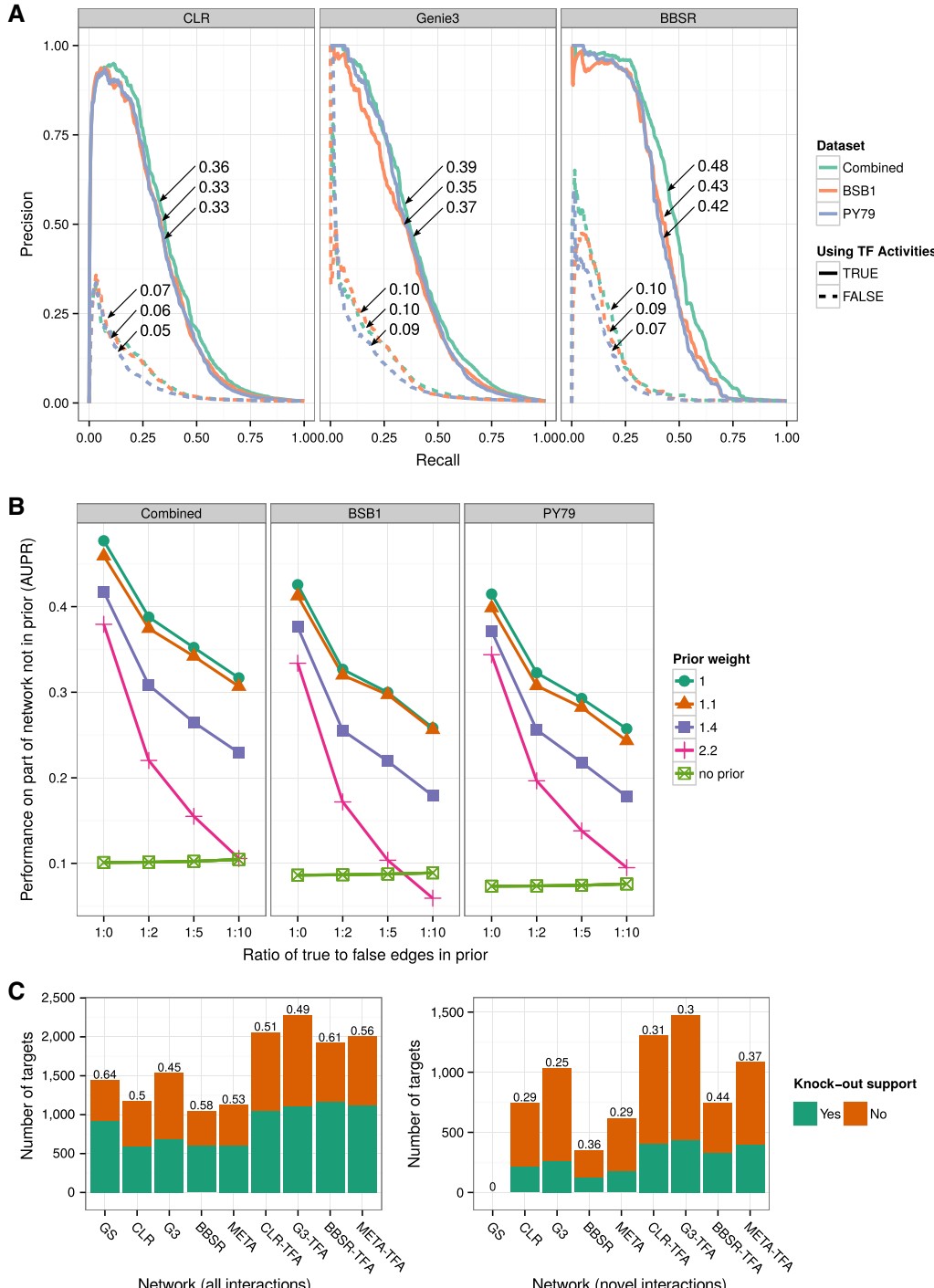

**Figure 3.  Performance of network inference methods when incorporating TFA.**

A   Precision–Recall plot of the confidence-ranked interaction networks using CLR, Genie3, and the Inferelator (no priors). Solid lines show performance using TFA. Dashed lines show performance when no TFA are used (when raw expression values for TFs are used as predictors). The numbers superimposed on each curve indicate the area under the curve.

B   Performance of BBSR-TFA (AUPR: area under precision recall curve) on the combined, BSB1 and PY79, networks in the presence of false prior information. 50% of the edges in the GS are used as true priors, and various amounts of random edges are added. Performance is evaluated on the leave-out set of interactions. Each point represents the median of five random samples of 50% of the GS set.

C   Support from KO data for the models predicted by BBSR-TFA, Genie3 (G3), CLR, and a consensus method (META) that rank combines the prediction of the three methods. Methods were used without and with TFA (TFA tag). The number on top of each bar indicates the proportion of evaluated interactions with KO support for the corresponding method. Left and right panels show the support for each method when all interactions (recovered priors and novel interactions) and only novel interactions are considered, respectively.

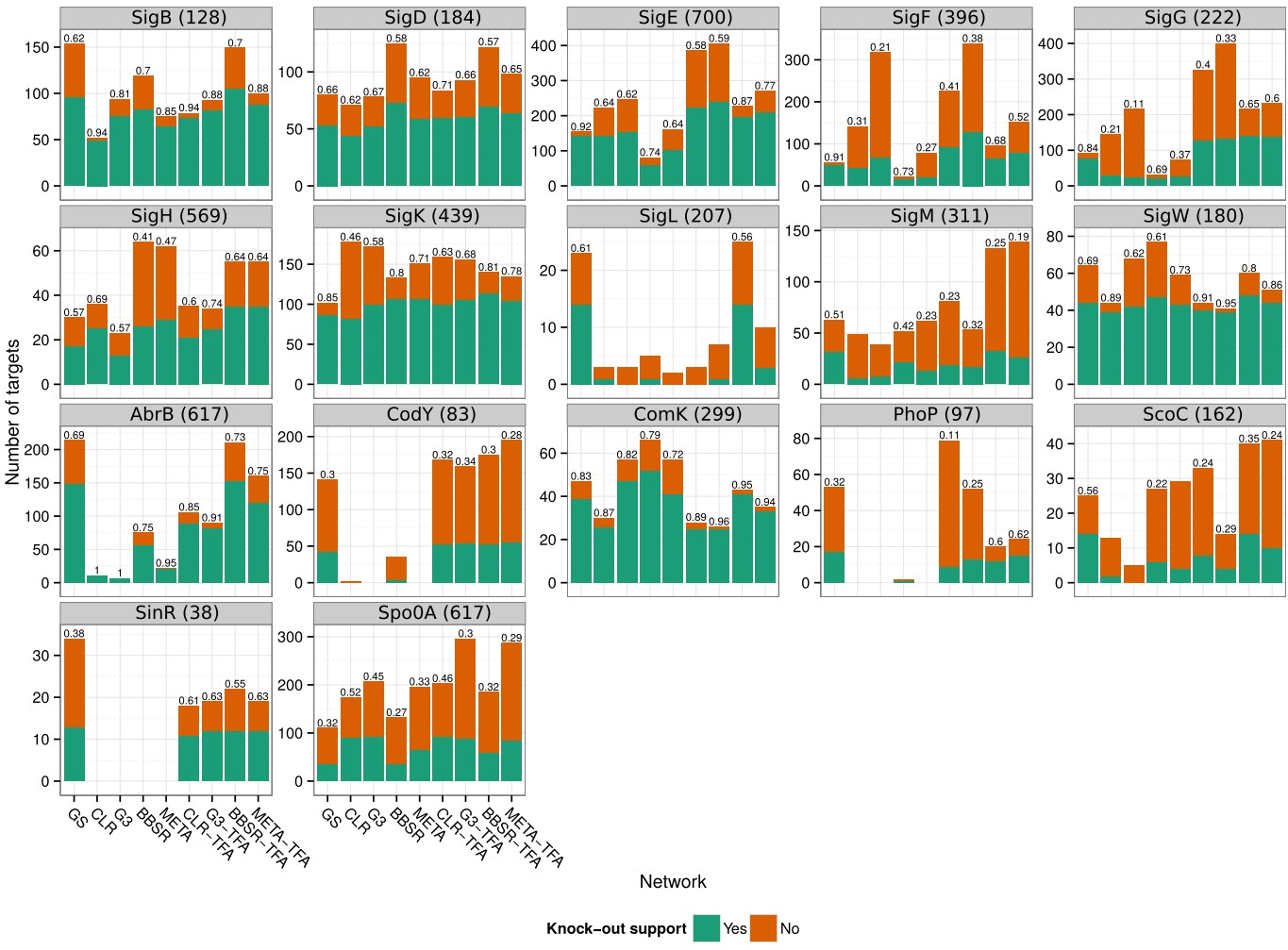

**Figure 4. Experimental support from KO data to the models predicted by BBSR, Genie3, and CLR.**
For each regulon with KO data, we assessed the proportion of predicted targets supported by the KO data. Results are presented for BBSR, Genie3 (G3), CLR, and a consensus method (META) that rank combines the prediction of the three methods. Methods were used without and with TFA (TFA tag). The number in parentheses next to the regulon's name indicates the total number of differentially transcribed genes in the corresponding KO data. The number on top of each bar indicates the proportion of evaluated interactions (recovered priors and novel interactions) supported by the KO data. This number is omitted if there was no significant (*P*-value ≥ 0.01) enrichment for differentially transcribed genes in the predicted targets

targets in the TF-specific evaluation network were considered true positives (i.e. supported by the KO data) if they were DT, while targets that were not DT were considered false positives (i.e. not supported by the KO data). Overall, we see that the KO support rate for the full set of tested predictions and novel predictions of BBSR-TFA is 0.61 and 0.44, respectively (Fig 3C). The median support rate per regulon was 0.64 for the full set and 0.49 for novel predictions (Fig 4 and Appendix Fig S4). We performed the same evaluation of the GENIE3 and CLR networks, as well as a consensus method (META) that rank combines the prediction of the three methods (as suggested by Marbach *et al*, 2012). The performance of all methods is shown in Fig 3C (analysis by regulon is shown in Fig 4 and Appendix Fig S4). We observed that BBSR-TFA outperforms the other methods with respect to the fraction of supported predictions, when all predictions (left panel) or only novel interactions (right panel) are considered, and all methods greatly benefited from using TF activities. The *Inferelator* (both BBSR and BBSR-TFA version) is

the most conservative method in predicting novel interactions, which is a result of including prior information in the model selection step. This also resulted in the lowest absolute and proportional number of unsupported novel predictions. The lower number of false positives is especially significant because it reduces the number of follow-up experiments required to confirm the predictions of the model.

**General features of the inferred TRN**

The inferred transcriptional network (Dataset EV4) contains 3,086 genes and predicts 4,516 interactions (2,258 novel interactions). Previously known interactions are recalled at high proportion (74% of the GS network is recovered and further supported by the new data, auto-regulation was not considered), while the global set of interactions is doubled (a direct result of selecting a 0.5 precision cutoff). Since transcriptional profiles instead of TFA

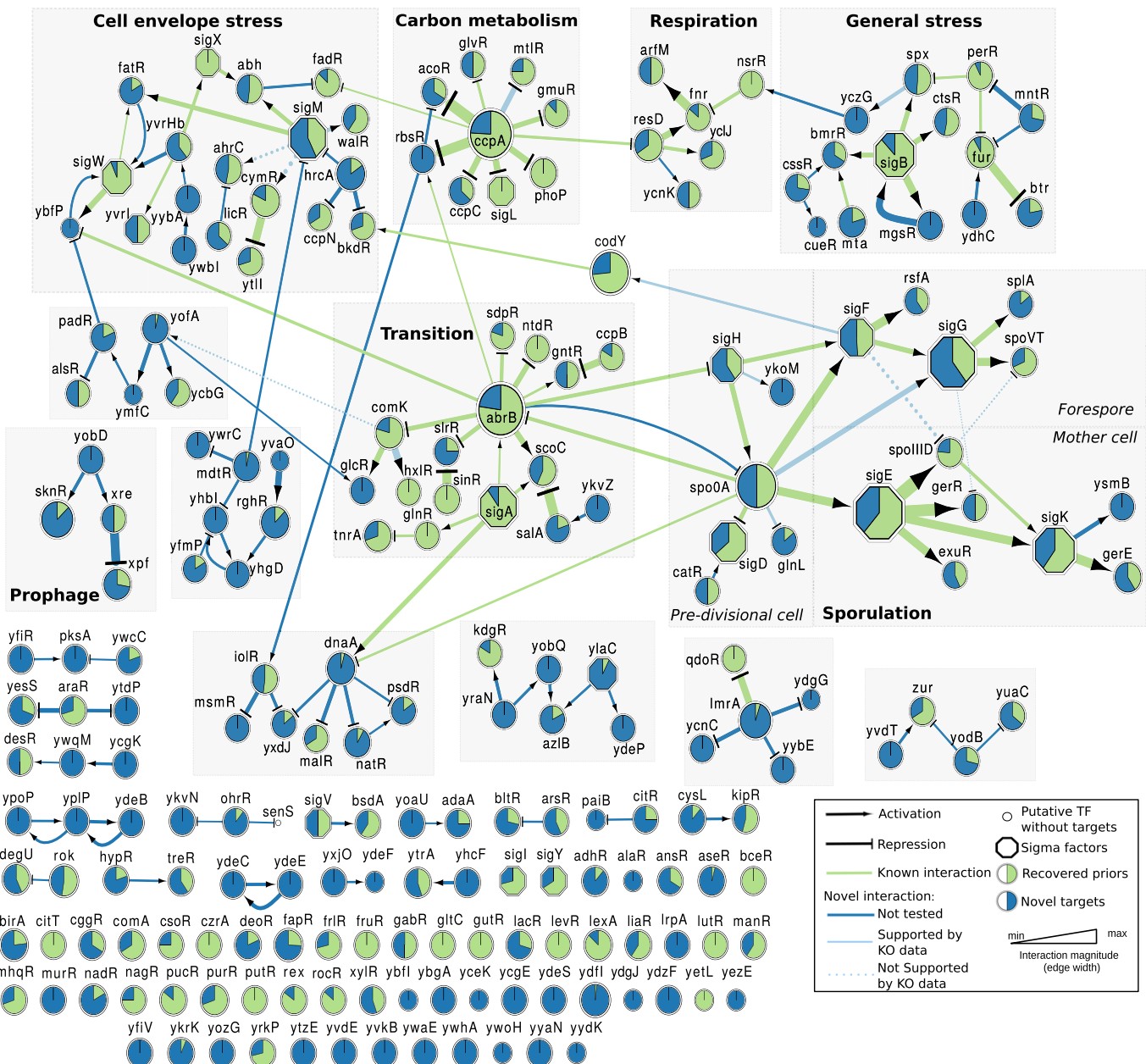

**Figure 5. Modular organization of the inferred TF network.**

Cytoscape view of the modular architecture of the inferred network, restricted to σ factors (octagons) and other transcription factors (circles). The size of each node reflects the total number of predicted targets. Modules are labeled based on the functional annotation of their members. Green edges are known interactions; blue edges are novel interactions. Pie chart within each node indicates the proportion of known members (i.e. present in the GS network, green) and novel members (blue) for each regulon. The corresponding Cytoscape file is provided in Dataset EV5.

were relied upon for TFs with no previously known targets, the average number of novel predictions was low for this group (4.7 per TF for TFs with no priors, compared to 22.5 per TF for TFs with more than ten priors). Similarly, ranking positions of novel interactions for TFs with less than six known targets were much lower than those for TFs with more than ten targets (Appendix Table S1).

From the model, we identify 11 global regulators with a minimum of 100 predicted target genes. This group includes σ$^A$ (major

σ factor) and six alternative σ factors (octagon symbols in Fig 5, see Dataset EV5 for the corresponding Cytoscape file, which includes both the full predicted network and the TF only network) distributed in three categories: (i) sporulation [σ$^E$ (early mother cell), σ$^G$ (late forespore), and σ$^K$ (late mother cell)]; (ii) other stress responses [σ$^B$ (general) and σ$^M$ (cell envelope)]; and (iii) motility (σ$^D$). The other global regulators are AbrB (transition to stationary phase); CcpA (carbon catabolite control); CodY (nitrogen and carbon metabolism); and Spo0A (sporulation).

To evaluate the degree of recall, we focused on 82 TFs with $\geq 5$ target genes in the prior network. Although the recovery rate was above 0.5 for most TFs, there were nine TFs with lower recovery (ExuR, GerE, GerR, SpoIIID, SpoVT, TnrA, YvrHb, $\sigma^A$, $\sigma^X$). Seven of these TFs have regulons that are either entirely comprised within or significantly overlapping with larger regulons (ExuR, GerE, GerR, SpoIIID, and SpoVT are targets of the sporulation $\sigma$ factors $\sigma^E$, $\sigma^G$, or $\sigma^K$ and participate in FFL network motifs, while the $\sigma^X$ regulon overlaps with that of $\sigma^M$, and the YvrHb regulon overlaps with the $\sigma^D$ and $\sigma^X$ regulons). The difficulty of predicting FFL motifs had been noted in previous network inference approaches (Marbach *et al*, 2012). Although this logic does not apply to TnrA, a regulator of nitrogen metabolism (Sonenshein, 2007), our dataset did not include experiments performed under conditions of nitrogen limitation. Recall of targets of the major sigma factor, $\sigma^A$, was also < 0.5, but this is a special case, considering that the number of genes regulated by $\sigma^A$ alone is likely to be quite small. In addition, genes with a coefficient of variation lower than 0.05 in either dataset were removed from the network inference procedure (see Materials and Methods).

**Experimental support for prediction of novel regulatory interactions in the inferred TRN**

The KO data (described above), the presence of putative binding sites for TFs in target promoters, and external validation data (primarily ChIP-seq and transcriptional profiling data published after compilation of the GS) were used to assess the experimental support for the regulons of alternative $\sigma$ factors and other global regulators (summarized in Table 1, full results in Dataset EV6). External validation data include experimental results for AbrB, CcpA, CodY, PhoP, Spx, TnrA, WalR, and Zur (Chumsakul *et al*, 2011; Marciniak *et al*, 2012; Rochat *et al*, 2012; Belitsky & Sonenshein, 2013; Salzberg *et al*, 2013, 2015; Brinsmade *et al*, 2014; Mirouze *et al*, 2015; Prestel *et al*, 2015). The proportion of recovered target operons with KO support is on average 0.83 for all $\sigma$ factors and 0.69 for the other TFs. We chose to count operons (i.e., transcription units), because of biological relevance, but results with gene counts do not differ significantly (compare Fig 4 to Table 1), except for long operons such as the 30 gene *fla-che* flagellum/chemotaxis operon. Overall, we found experimental support for 1,289 TF–gene interactions (out of 1,841 tested) in transcriptional profiling data with KO strains, including 391 (out of 635) novel interactions. By sequence analysis, we also identified putative binding sites in 71% of the operons predicted as novel targets ($\sigma^B$, $\sigma^D$, $\sigma^E$, $\sigma^G$, $\sigma^H$, $\sigma^K$, $\sigma^W$, CcpA, and CodY regulons; Appendix Fig S5). We also included the predictions from Eichenberger *et al* (2004) and Nicolas *et al* (2012) in the analysis of $\sigma$ factor binding sites, Marciniak *et al* (2012) for *cre* sites (CcpA binding), and Mirouze *et al* (2015) and Leyn *et al* (2013) for TnrA binding sites (Dataset EV6). In total, there were 754 interactions (out of 1,258 tested) supported by both KO data and sequence analysis. Lastly, due to the presence of putative TF binding sites matching known consensus binding sequences, we obtained supporting evidence for 58 interactions in the CymR, Fur, LexA IolR, and Zur regulons.

Because not every interaction in the GS was recalled in our model, we checked whether the target genes in these interactions were DT in the corresponding KO experiment. The global rate of differential transcription for these missing interactions (i.e. those present in the GS network but absent in our model) was 0.38, suggesting that at least some of the interactions in the GS may be either inaccurate or strictly dependent on strain background and/or specific experimental conditions. In any event, this number is significantly lower than the support rate for the full predicted network (0.7), the set of interactions recovered from the GS (0.74), or the set of novel interactions (0.62).

We also analyzed the top 500 novel predicted interactions in the final combined network (ranked by the associated confidence score, Dataset EV7). The top 500 novel predictions include 483 target genes and 91 TFs. Forty-one of these interactions have been validated by external sources since compilation of the GS, and four have been validated in the current study using GFP fusions. Seventy-six of the remaining interactions can also be validated on the grounds that the genes involved belong to operons that include previously known targets. This applies to many short genes that were added to the genome after re-annotation of the *B. subtilis* 168 sequence (Barbe *et al*, 2009). These genes were absent from microarrays generated from the original genome annotation (Kunst *et al*, 1997) and would have been missed in transcriptional profiling experiments conducted prior to the year 2010. Considering the remaining 378 interactions, transcriptional profiling data were available (from this and previous studies) for 210 interactions. We found that 153 out of these 210 interactions (i.e. 73%) were experimentally supported by transcriptional profiling data (*P*-value < 0.01 and/or external validation). In parallel, we performed a search and/or collected information from previous studies for the presence of putative binding sites for TFs involved in 193 putative interactions. A sequence motif compatible with a previously reported consensus binding site was identified in the corresponding promoter sequences for 136 out of these 193 interactions (70%), thus providing additional evidence for these predictions. In total, there were 144 interactions for which both KO and motif data were available. Out of these 144 putative interactions, 120 (83%) were supported by both (in addition to the 122 predictions that we considered already validated by external sources). These findings suggest high prediction accuracy for the top 500 predictions (when ranked by confidence score).

**Architecture of the inferred TRN**

We clustered the 215 TFs in our model to explore the topology of the inferred TRN. Seven of thirteen modules (defined as clusters with $\geq 4$ TFs) were enriched in specific processes (Fig 5, Dataset EV5 for the Cytoscape file). From top to bottom, and left to right, in Fig 5: (i) Cell envelope stress response: with four $\sigma$ factors ($\sigma^M$, $\sigma^W$, $\sigma^X$, and YvrI), three TFs involved in the maintenance of the integrity of the cell wall and plasma membrane (FatR, WalR, and YvrHb) and two TFs (CymR and YtlI) involved in sulfur metabolism (however, the predicted interaction between $\sigma^M$ and CymR is not supported by the *sigM* KO data). (ii) Carbon metabolism: with CcpA, the global regulator of carbon catabolite control and $\sigma^L$. (iii) Cellular respiration: with ResD as main TF, and ArfM and Fnr as regulators of anaerobic genes. (iv) General stress response: with $\sigma^B$, regulators of the oxidative stress response (Spx and PerR) and metal homeostasis (PerR, Fur, and MntR). (v) Prophage: with Xre and Xpf

**Table 1. Support provided by transcriptional profiling experiments with KO strains.**

| TF | Priors[a] | Priors predicted as targets[a] | Recovery | Recovered priors supported | Accuracy (priors) | Novel predicted targets[a] | Novel targets supported[a] | Accuracy (novel targets) | Predictions (recovered + novel targets[a]) | Supported (recovered + novel targets[a]) | Accuracy (total) |
|---|---|---|---|---|---|---|---|---|---|---|---|
| $\sigma^B$ | 103 | 96 | 0.93 | 70 | 0.73 | 14 | 8 | 0.57 | 110 | 78 | 0.71 |
| $\sigma^D$ | 29 | 27 | 0.93 | 24 | 0.89 | 20 | 6 | 0.3 | 47 | 30 | 0.64 |
| $\sigma^E$ | 90 | 85 | 0.94 | 80 (82[b]) | 0.94 (0.98[b]) | 61 | 52 (53[b]) | 0.85 (0.87[b]) | 146 | 132 (133[b]) | 0.90 (0.91[b]) |
| $\sigma^F$ | 39 | 33 | 0.85 | 32 | 0.97 | 25 | 21 | 0.84 | 58 | 53 | 0.91 |
| $\sigma^G$ | 64 | 58 | 0.91 | 53 (58[c]) | 0.91 (1.0[c]) | 69 | 55 (67[c]) | 0.8 (0.97[c]) | 127 | 108 (125[c]) | 0.85 (0.98[c]) |
| $\sigma^H$ | 22 | 18 | 0.82 | 13 | 0.72 | 31 | 19 | 0.61 | 49 | 32 | 0.65 |
| $\sigma^K$ | 58 | 54 | 0.93 | 47 (53[d]) | 0.87 (0.98[d]) | 50 | 31 (47[d]) | 0.62 (0.94[d]) | 104 | 78 (100[d]) | 0.75 (0.96[d]) |
| $\sigma^L$ | 6 | 6 | 1.0 | 4 | 0.67 | 0 | 0 | NA | 6 | 4 | 0.67 |
| $\sigma^M$ | 31 | 26 | 0.84 | 19 | 0.73 | 48 | 2 | 0.04 | 74 | 21 | 0.28 |
| $\sigma^W$ | 33 | 28 | 0.85 | 23 | 0.82 | 1 | 1 | 1.0 | 29 | 24 | 0.83 |
| AbrB-repr[e] | 101 | 77 | 0.76 | 54 | 0.7 | 26 | 17 | 0.65 | 103 | 71 | 0.69 |
| CcpA-repr[f] | 72 | 51 | 0.7 | 38 | 0.75 | 20 | 9 | 0.45 | 71 | 47 | 0.66 |
| CodY-repr | 37 | 30 | 0.81 | 14 (26[g]) | 0.47 (0.87[g]) | 17 | 8 (12[g]) | 0.47 (0.71[g]) | 47 | 22 (38[g]) | 0.47 (0.81[g]) |
| ComK-activ | 16 | 12 | 0.75 | 12 | 1.0 | 7 | 6 | 0.86 | 19 | 18 | 0.95 |
| PhoP-activ | 18 | 11 | 0.61 | 3 (7[h]) | 0.3 (0.64[h]) | 0 | NA | NA | 11 | 3 (7[h]) | 0.3 (0.64[h]) |
| ScoC-repr | 10 | 6 | 0.60 | 3 | 0.5 | 9 | 2 | 0.2 | 15 | 5 | 0.33 |
| SinR-repr | 13 | 4 | 0.31 | 3 | 0.75 | 0 | NA | NA | 4 | 3 | 0.75 |
| Spo0A-activ | 24 | 14 | 0.58 | 8 | 0.57 | 28 | 10 | 0.36 | 42 | 18 | 0.43 |
| Spo0A-repr | 21 | 14 | 0.67 | 6 | 0.43 | 24 | 4 | 0.17 | 38 | 10 | 0.26 |
| Spx[i] | 11 | 10 | 0.91 | 5 | 0.5 | 17 | 7 | 0.41 | 27 | 12 | 0.44 |
| TnrA-activ[j] | 12 | 7 | 0.58 | 7 | 1.0 | 0 | NA | NA | 7 | 7 | 1.0 |
| TnrA-repr[j] | 11 | 4 | 0.36 | 4 | 1.0 | 8 | 0 | 0 | 12 | 4 | 0.33 |
| WalR[k] | 8 | 7 | 0.88 | 5 | 0.71 | 7 | 0 | 0 | 14 | 5 | 0.36 |
| Zur[l] | 4 | 4 | 1.0 | 4 | 1.0 | 3 | 1 | 0.33 | 7 | 4 | 0.57 |

This is a summary of the data presented in Dataset EV6 (Analysis by Regulon). Operons are considered to be differentially transcribed when at least half of the genes in the operon have a *P*-value $\leq 0.01$ in transcriptional profiling experiments (WT versus KO).
[a]Numbers refer to operons (i.e. transcription units, not individual genes).
[b]Includes operons supported for SpoIIID dependency ($\sigma^E$ and SpoIIID form a FFL).
[c]Includes operons supported for SpoVT dependency ($\sigma^G$ and SpoVT form a FFL).
[d]Includes operons supported for GerE dependency ($\sigma^K$ and GerE form a FFL).
[e]For AbrB dependency, we consider differential transcription in a *spo0A* gene deletion strain versus a wild-type strain.
[f]Based on data from Marciniak *et al* (2012).
[g]Based on data from Belitsky and Sonenshein (2013) and Brinsmade *et al* (2014).
[h]Based on data from Salzberg *et al* (2015).
[i]Based on data from Rochat *et al* (2012).
[j]Based on data from Mirouze *et al* (2015).
[k]Based on data from Salzberg *et al* (2013).
[l]Based on data from Prestel *et al* (2015).

regulating expression of genes in the PBSX prophage and SknR in the *skin* element. (vi) Transition from exponential growth to stationary phase: This cluster represents the core of the TF network, because it is connected to most of the other modules. It contains TFs directly involved in the regulation of the transition phase (AbrB and ScoC) and many TFs required for various cell differentiation processes, such as biofilm formation (SinR and SlrR), cannibalism (SdpR), and competence (ComK); it also includes TFs involved in nitrogen metabolism (TnrA and GlnR). (vii) Sporulation: This module is composed of three sub-networks: (i) pre-divisional cell (control of gene expression prior to asymmetric division of the sporulating cell), this sub-network is organized around Spo0A and $\sigma^H$ (both of which connected to

the transition module via AbrB), and it also contains the motility σ factor $\sigma^D$ (i.e. cell motility is turned off in sporulating cells); (ii) forespore, with $\sigma^F$ and $\sigma^G$, includes an intriguing novel connection between $\sigma^F$ and co*dY* (considering that CodY is a repressor of several biosynthetic processes, its compartmentalized expression during sporulation could contribute to shutting down metabolic activity in the forespore); and (iii) mother cell, with $\sigma^E$ and $\sigma^K$. The mother cell sub-network includes a previously uncharacterized putative regulator, YsmB. It should be noted that *ysmB* is located immediately downstream of *gerE*, which encodes a well-known regulator of mother cell gene expression, and the two genes (along with *racE*) constitute an operon (some of the predicted targets of YsmB are differentially transcribed in the

*gerE* KO experiment, but it is unclear whether these genes are regulated by GerE, YsmB, or both (see Dataset EV6, σ$^K$ regulon). There were six clusters that we did not annotate due to lack of functional enrichment. These clusters are composed primarily of novel regulatory interactions that have not been assessed experimentally. Functionally, many of the genes found in these clusters are annotated as unknown, providing resistance to toxic compounds/antibiotics or involved in carbon metabolism.

## The inferred TRN provides novel functional insights even for well-characterized pathways

We calculated how many genes involved in defined cellular processes were paired with TFs in the prior and inferred networks (Appendix Table S2). Many genes involved in extensively studied processes that were missing in the prior network (e.g. about a quarter of the genes annotated as "sporulation", "stress response," and "exponential and early post-exponential lifestyles") have now been linked to specific regulators in the inferred TRN. The "exponential and early post-exponential lifestyles" category is composed of three very well-studied processes: "motility and chemotaxis", "biofilm formation," and "competence". Specifically, our model increases the proportion of genes with at least one candidate TF in all categories (from a median proportion by category of 0.51 in the prior network to 0.72 in the inferred network; Appendix Table S2). In the inferred TRN, sporulation (528 genes) is the category with the highest proportion of regulatory hypotheses. Our TRN model can serve as a guide for the analysis of many uncharacterized genes, because 521 genes of unknown function (out of 872) are now paired with TFs. As an illustration, we discuss below the identification of new targets of σ$^K$, while in the appendix, we provide information on new targets of ComK (Appendix Fig S6) and CodY (Appendix Fig S7).

### Sporulation (σ$^K$ regulon)
Most predicted targets of sporulation σ factors were experimentally supported in transcriptional profiling experiments with corresponding KO strains (Table 1, Dataset EV6). The prediction accuracy for sporulation σ factors ranges from 0.75 (σ$^K$) to 0.91 (σ$^F$). In the Appendix, we present fluorescence microscopy data to validate several novel sporulation genes (*yetF*, *ykzQ*, and *ykoST*; Appendix Fig S8). Because of our interest in spore coat assembly, we analyzed novel targets of σ$^K$ and GerE (McKenney *et al*, 2013). These TFs form two FFL motifs, where σ$^K$ is the first regulator, and GerE is either an activator in a coherent FFL or a repressor in

an incoherent FFL (Eichenberger *et al*, 2004). In the sporulation regulatory cascade, σ$^K$ is the last σ factor to be activated. Our model predicts that 161 genes (104 operons) are controlled by σ$^K$, including 64 novel target genes (50 operons). One of these novel genes is *ytdA* (Fig 6A). We confirmed that *ytdA* transcription is σ$^K$-dependent (Fig 6B, left), a result consistent with the presence of a putative σ$^K$ binding site in the *ytdA* promoter (Fig 6B, right). We also showed that YtdA-GFP displays the typical localization pattern of a spore coat protein (Fig 6C). Lastly, YtdA-GFP is still produced in the absence of GerE; albeit with a disrupted localization pattern, suggesting that YtdA-GFP recruitment to the spore surface is dependent on GerE-controlled sporulation genes.

## Characterization of genes involved in spore polysaccharide synthesis

The novel σ$^K$-dependent gene, *ytdA*, encodes a putative UTP-glucose-1-phosphate uridylyltransferase with two paralogs, *spsI* and *yfnH*, also under σ$^K$ control (Eichenberger *et al*, 2004) (Fig 6A).The last four genes in the *sps* (spore polysaccharide synthesis) operon are necessary for synthesis of rhamnose (Plata *et al*, 2012), a carbohydrate present on the *B. subtilis* spore surface (Wunschel *et al*, 1995). To confirm that YtdA is a coat protein, we showed that YtdA-GFP localization was disrupted in the absence of CotE, a protein required for assembly of the outer coat (Fig 6C, top); however, YtdA-GFP localization was not affected in a *cotXYZ* mutant lacking the outermost coat layer, the crust (McKenney *et al*, 2010). SpsI-GFP had similar requirements (Fig 6C, bottom), as did SpsK-CFP, albeit transiently (Fig 6D, single cap of fluorescence on the mother cell-proximal side of the forespore). By contrast, SpsJ-YFP and its paralog, YtcB-YFP, did not localize to the forespore surface (Fig 6D). The third paralog, YfnH-YFP, exhibited a two-step localization pattern (Fig 6E): first as a diffuse signal (hour 7) and, by hour 8, as bright foci in the mother cell cytoplasm. Localization of YfnH-YFP foci was reminiscent of SpsM, another previously characterized spore polysaccharide synthesis protein (Abe *et al*, 2014). In dual labeling experiments, we showed that YfnH-YFP and SpsM-CFP co-localized (Fig 6E, white arrows). In total, our fluorescence microscopy data suggest that one pathway of spore polysaccharide synthesis (involving SpsI, SpsK, and YtdA) occurs directly on the spore coat, while another one may be going on in the mother cell cytoplasm (with SpsM and YfnH).

Next, we generated a triple gene deletion mutant of *spsI*, *ytdA*, and *yfnH*. Spore adhesion assays (Fig 6F and G) showed that triply mutant spores more strongly adhere to glass tubes than wild-type

**Figure 6. Functional analysis of spore polysaccharide synthesis genes.**

A Genomic organization of the *ytdA*, *yfnH*, and *spsI* gene regions (*ytdA* is a paralog of *yfnH* and *spsI*). Putative gene functions are color-coded. β score for each prediction is indicated in parenthesis.

B Left: Fluorescence microscopy images for the YtdA-GFP fusion in sporulating cells in the indicated mutant backgrounds. Except where indicated otherwise, images were collected for sporulating cells at hour 6 after suspension in Sterlini–Mandelstam medium at 37°C. Right: Possible binding site for σ$^K$ in the *ytdA* promoter. The consensus binding sequence for σ$^K$ is also indicated (M is A or C).

C Spore coat localization of YtdA-GFP and SpsI-GFP is dependent on *cotE* and independent of *cotXYZ*.

D Subcellular localization of YtcB-YFP, SpsJ-YFP, and SpsK-CFP during sporulation.

E Time course and dual labeling analysis of YfnH-YFP and SpsM-CFP during sporulation.

F Spore adhesion to glass: A *spsI* deletion mutant and a *spsI ytdA* double mutant display strong adhesion.

G Spore adhesion to hydrocarbons (hexadecane): A *spsI* deletion mutant and a *spsI ytdA* double mutant display strong adhesion. A *spsI yfnH* double mutant and a *spsI yfnH ytdA* triple mutant display intermediate adhesion. Error bars represent the standard deviation for three independent experiments.

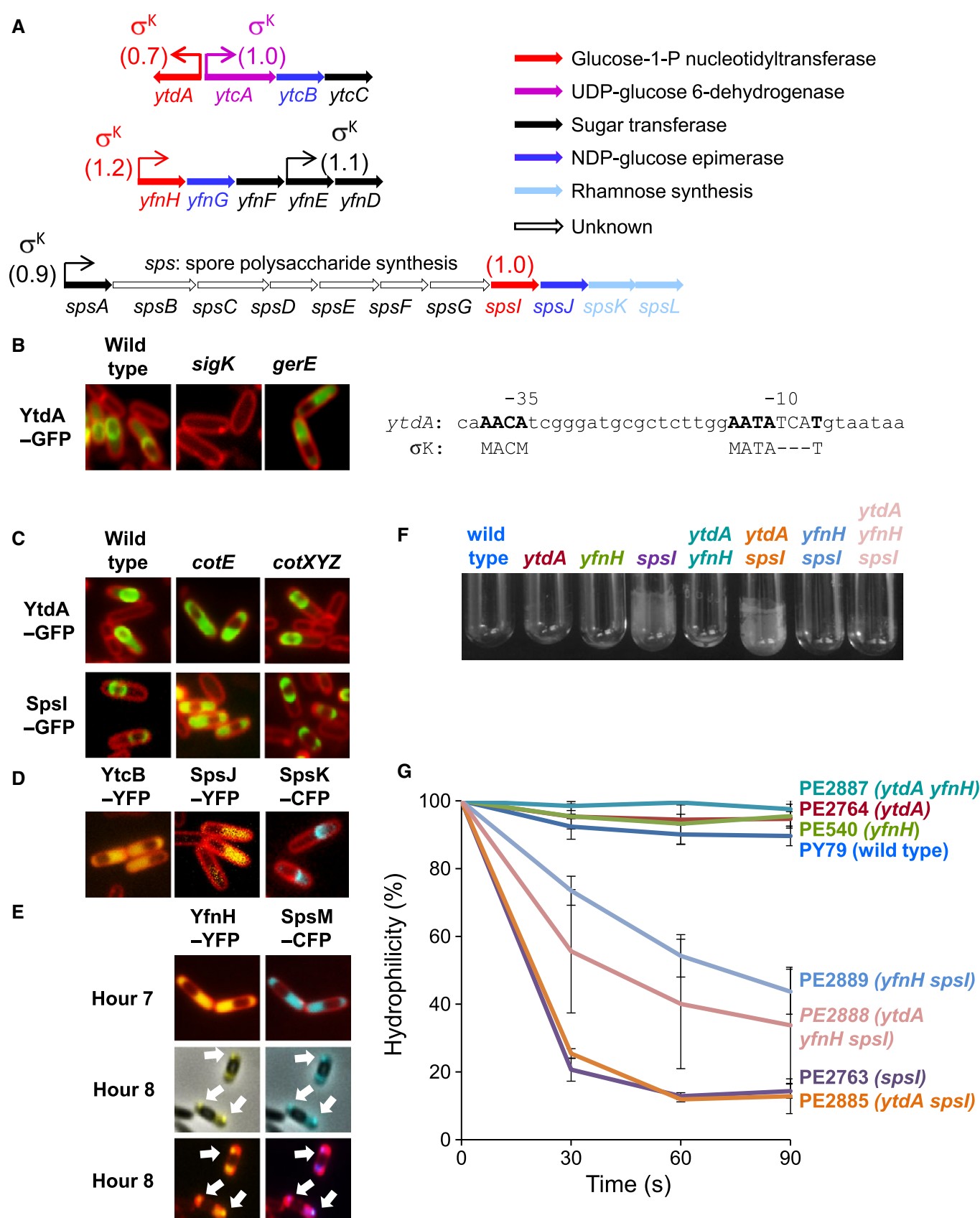

**Figure 6.**

spores (PY79), suggesting that the spore surface is more hydrophilic in the presence of polysaccharides, thus favoring spore dispersal in water (Abe *et al*, 2014). The main contributor to this property is SpsI, because a *spsI* mutant showed the strongest adhesion phenotype. In summary, deletions of spore polysaccharide synthesis genes (especially *spsI*) result in significant modifications of spore surface properties.

## Conclusions

Our inferred model of the *B. subtilis* TRN is a significant improvement over the initial GS network and previous TRN models. Our inferred TRN recalls a high proportion of known interactions and simultaneously adds 2,258 putative interactions. Our methods were tested in a variety of contexts (using real and synthetic data) and shown to tolerate error in the GS (in excess of that expected here). Our predictions are associated with error estimates that can be used to guide biologists using the model. Importantly, we obtained further experimental support for 391 (out of 635 evaluated) novel regulatory edges (representing 17% of all novel interactions), demonstrating the high accuracy (62%) of our predictions. The number of *B. subtilis* genes devoid of regulatory hypotheses has thus been significantly reduced (not just for genes of unknown function, but also for genes in well-known pathways such as sporulation, other stress responses, metabolism, and competence, Appendix Table S2).

The new network model exhibits more connections spanning function and cell process boundaries, suggesting previously unsuspected links between cellular processes. Our model could, however, be further improved by the addition of data for conditions that have remained untested. In future network inference attempts, the inclusion of complementary data types should be prioritized, especially genomewide binding assays (for poorly characterized TFs) and proteomics (to characterize post-transcriptional regulatory events). The most important conclusion from our work is that incorporating TFA critically improves the predictive performance of network inference approaches, while maintaining a high tolerance to error in the methods used to generate these structure priors. Our results further suggest that estimating TFA increases the ability to distinguish true transcriptional interactions from random correlations, in particular for TFs that are activated by post-translational modifications and/or require co-factors. Moreover, using TFA reduces the number of predictions with incorrect sign. A remaining limitation is that we make fewer confident predictions for regulators with few (or no) known targets, because TFA estimation is less accurate. Overall, the strategy delineated here can be applied to other bacteria and eukaryotic cells as long as a minimal set of priors and large transcriptional datasets are available.

# Materials and Methods

## Media and growth conditions, strains, plasmids, and primers

A detailed description of culture media and growth conditions is provided in the Appendix. All *B. subtilis* strains used in this study were derivatives of the wild-type strain PY79 (Table EV1). Strains expressing fusions to fluorescent proteins were generated as previously described (McKenney *et al*, 2010).

## Collection of transcriptomic datasets

### RNA isolation, cDNA synthesis, labeling, and hybridization to microarrays

RNA was extracted for a total of 403 samples in 38 separate experimental designs (Table EV2), converted to cDNA, fluorescently labeled, and hybridized to microarrays (PY79 dataset, GEO accession number GSE67023). These procedures have been described before (Cozy *et al*, 2012). The previously published BSB1 dataset is accessible at GEO with accession number GSE27219 (Nicolas *et al*, 2012).

### Microarray design

Agilent's eArray software was used to design 60-mer probes (features) for all annotated protein-coding genes (three probes per gene) from the *B. subtilis* 168 genome (Barbe *et al*, 2009). The "b.subt-final-sense-3probes-july29" array was obtained from Agilent (GEO accession number GPL15179).

### Processing microarray data

For the PY79 dataset, we performed the following steps separately for each channel: (i) take the median of all probes of the same gene; (ii) log$_2$-normalize the intensities; (iii) normalize between arrays using the cyclic loess method; and (iv) average replicates of common reference conditions.

## Inferring the regulatory network

### Gene filter

For network inference, we only considered genes with a coefficient of variation > 0.05 in either dataset. For consistency, we also removed genes that were unique to one of the two strains or microarray platforms [the list of genes considered for network inference is provided in Dataset EV1 (Excel sheet 2)].

### Gold standard

A set of experimentally validated transcriptional interactions was downloaded from SubtiWiki (Michna *et al*, 2014), while a list of σ$^A$-controlled genes was obtained from the study by Helmann (1995). Genes filtered out due to a lack of expression variance were also removed from the GS. The final GS includes 3,040 interactions involving 1,874 genes (Dataset EV2).

### Estimating transcription factor activities

Let $X$ be the matrix of gene expression values, where rows are genes and columns represent experiments/samples. Let $P$ be a matrix of known regulatory relationships between transcription factors (columns) and target genes (rows). The entries in the prior matrix ($P$, derived directly from the GS set described above) are members of the set $\{-1, 0, 1\}$. We set $P_{i,j}$ to zero if there is no known regulatory interaction between transcription factor (TF) $j$ and gene $i$, to minus one if TF $j$ is known to repress gene $i$, and to one if TF $j$ is known to activate gene $i$. Auto-regulatory interactions are always set to zero in $P$. Estimation of TF activities is then based on the following model (Liao *et al*, 2003; Fu *et al*, 2011):

$$X_{i,j} = \sum_{k \in TFs} P_{i,k} A_{k,j}$$

where the expression of gene $i$ in sample $j$ can be written as the weighted sum of connected TF activities $A$ (with the key distinction/modification that we load activators and repressors into P separately as 1 and −1, respectively). In matrix notation, this can be written as X = $PA$, which we solve for the unknown TF activities $A$. This is an overdetermined system, but we can find $\hat{A}$ which minimizes $||P\hat{A} - X||^2$ using the pseudoinverse of $P$. Special treatment is given to time series experiments, with the modified model:

$$X_{i,t_n+\frac{\tau}{2}} = \sum_{k \in TFs} P_{i,k} A_{k,t_n}$$

where the expression of gene $i$ at time $t_n + \tau/2$ is used to inform the TF activities at time $t_n$. Here, $\tau$ is the time shift between TF expression and target expression used when inferring regulatory relationships (see next section). Here, we use a smaller time shift of $\tau/2$, because changes in TF activities should be temporarily closer to target gene expression changes. If there is no expression measurement at time $t_n + \tau/2$, we use linear interpolation to fit the values. In cases where there are no known targets for a TF, we cannot estimate its activity profile, and use the observed transcription as a proxy instead.

### Inferring regulatory relationships

The main input to the network inference procedure is the expression data X, the estimated transcription factor (TF) activity $\hat{A}$, and the known regulatory relationships encoded in the matrix $P$. The core model is based on the assumption that the expression of a gene $i$ at condition $j$ can be written as linear combination of the activities of the TFs regulating it. Specifically, in the case of steady-state measurements, we assume

$$X_{i,j} = \sum_{k \in TFs} \beta_{i,k} \hat{A}_{k,j} \tag{1.1}$$

For time series data, we explicitly model a time shift between the target gene expression response and the TF activities:

$$X_{i,t_n} = \sum_{k \in TFs} \beta_{i,k} \hat{A}_{k,t_n-\tau} \tag{1.2}$$

Here, we are modeling the expression of gene $i$ at time $t_n$ as the sum of activities at time $t_n - \tau$, where $t_n$ is the time of the $n^{\text{th}}$ measurement in the time series and $\tau = 15$ min is the desired time shift. In cases where we do not have measurements for $\hat{A}_{k,t_n-\tau}$, we use linear interpolation to add missing data points.

The goal of our inference procedure is to find a sparse solution to β, that is, a solution where most entries are zero. The left hand sides of eqns (1.1) and (1.2) are concatenated as response, while the right hand sides are concatenated as design variables. We use our previously described method Bayesian Best Subset Regression (BBSR) (Greenfield *et al*, 2013) to solve for β. With BBSR, we compute all possible regression models for a given gene corresponding to the inclusion and exclusion of each potential predictor. For a given target gene $i$, potential predictors are those TFs that have a known regulatory effect on $i$, and the ten TFs with highest time-lagged CLR (Greenfield *et al*, 2010; Madar *et al*, 2010). Prior knowledge is incorporated by using a modification of Zellner's g-prior (Zellner, 1983) to include subjective information on the regression parameters. A

g-prior equal to 1.1 was used for the combined network described in this study. Sparsity of our solution is enforced by a model selection step based on the Bayesian information criterion (BIC) (Schwarz, 1978).

### Ranking interactions and bootstrapping

After model selection is carried out, the output is a matrix of dynamical parameters β, where each entry corresponds to the direction (i.e. activation or repression) and strength (i.e. magnitude) of a regulatory interaction. These parameters can be used to predict the response of the system to new perturbations. In order to rank predicted regulatory interactions by confidence, we take into account the overall performance of the model for gene $i$ and the proportion of variance explained by each $\beta_{i,k}$ (for details see Greenfield *et al*, 2013). To further improve inference and become more robust against over-fitting and sampling errors, we employ a bootstrapping strategy. We resample the input conditions with replacement and run model selection on the new data set. This procedure is repeated 100 times, and the resulting lists of interactions are rank combined to a final ranked list as in the study by Marbach *et al* (2010). The final confidence score of an interaction is defined as the mean of the normalized rank across all bootstraps, where at each bootstrap, interactions are ranked by variance explained. The final network consists of the N top ranked interactions, where N is the maximum value so that at least 50% of the gold standard interactions are recovered.

### Combining the PY79 and BSB1 networks

We inferred the regulatory networks of PY79 and BSB1 independently. Then, we rank combined all 200 networks (100 for each strain), where the ranks of interactions are based on the confidence calculated in the previous step. Similarly, we averaged the β scores for each interaction. For any given downstream analysis, we define a confidence threshold for selecting interactions to include in the model (a threshold where precision is calibrated using AUPR curves as illustrated in Fig 3A).

### Other inference methods

In addition to the network inference described above, we built networks using CLR (Faith *et al*, 2007) and Genie3 (Huynh-Thu *et al*, 2010).

## Network exploration and validation experiments

### Network visualization

The combined network was visualized using Cytoscape v. 2.8 (Smoot *et al*, 2011).

### Computational search for DNA sequence motifs

The MEME suite (Bailey *et al*, 2009) was used for identifying DNA sequence motifs characteristic of binding by individual TFs. We used the DNA sequence up to 200 bp upstream of the translational start site for each operon predicted as target.

### Differential gene expression analysis

We compared gene expression profiles of wild-type (WT) and the respective mutant strains (KO) using Bayesian *t*-tests with the

Cyber-T tool (Baldi & Long, 2001). We followed the authors' recommendation to keep the number of replicates plus the confidence parameter equal to ten. Genes with *P*-values equal or smaller than 0.01 were considered differentially transcribed (DT).

*Fluorescence microscopy*

Microscopy experiments were performed as previously described (McKenney *et al*, 2010).

*Glass tube adhesion assays*

The procedure has been described in the study by Abe *et al* (2014).

*Spore adhesion to hydrocarbons assay*

The hydrophobicity of the spores was tested as described by Faille *et al* (2010) with the following modifications. Purified spores are re-suspended in PBS to a final $OD_{600}$ of 0.4–0.6. Three milliliters of each sample are set out for three separate exposure experiments. Five hundred microliters of hexadecane are added to each sample, and they are vortexed gently at the different hexadecane exposure times of 30, 60 and 90 s. The samples are then left to settle for 30 min, allowing for hydrophobic spores to travel to the hexadecane layer. The $OD_{600}$ is then measured for the aqueous phase of each sample. Percent hydrophilicity is calculated using: ($OD_{600}$ at exposure time/$OD_{600}$ initial) × 100.

## Data availability

The new gene transcription data have been deposited in GEO, accession number GSE67023.

## Software availability

The Inferelator software is available at http://bonneaulab.bio.nyu.edu/networks.html.

**Expanded View** for this article is available online.

## Acknowledgements

We thank David Dubnau, Carol Gross, John Helmann, Adriano Henriques, Dan Kearns, Rich Losick, and Daniel Ziegler for providing strains. This work was supported by NIH grant GM092616 to RB, PE, and DZR, NIH grant GM081571 to PE, and the Simons Foundation.

## Author contributions

MLAO, CH, ARB, TC, AG, BS, SNB, MG, BL, TK, FS, JC, CDAR, TS, DZR, AD, RB, and PE designed the experiments; MLAO, CH, ARB, TC, BS, SNB, MG, BL, FS, JC, CDAR, and AD performed the experiments; MLAO, CH, ARB, TC, AG, BS, SNB, MG, BL, TK, FS, JC, CDAR, TS, DZR, AD, RB, and PE analyzed the data; and MLAO, CH, ARB, RB, and PE wrote the paper.

## Conflict of interest

The authors declare that they have no conflict of interest.

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
