## [Review Process File · Molecular Systems Biology]

An Experimentally Supported Model of the *Bacillus subtilis* Global Transcriptional Regulatory Network

Mario L Arrieta-Ortiz, Christoph Hafemeister, Ashley Rose Bate, Timothy Chu, Alex Greenfield, Bentley Shuster, Samantha N Barry, Matthew Gallitto, Brian Liu, Thadeous Kacmarczyk, Francis Santoriello, Jie Chen, Christopher D.A. Rodrigues, Tsutomu Sato, David Z Rudner, Adam Driks, Richard Bonneau and Patrick Eichenberger

Corresponding author: Patrick Eichenberger, New York University

Review timeline:

Submission date:	15 April 2015
Editorial Decision:	26 May 2015
Revision received:	08 September 2015
Editorial Decision:	08 October 2015
Revision received:	15 October 2015
Accepted:	21 October 2015

Editor: Maria Polychronidou

Transaction Report:

1st Editorial Decision

26 May 2015

Thank you again for submitting your work to Molecular Systems Biology. We have now heard back from the three referees who agreed to evaluate your manuscript. As you will see from the reports below, the referees acknowledge that the study seems potentially interesting. However, they raise a series of concerns, which should be carefully addressed in a revision of the manuscript.

Without repeating all the points listed below, some of the more fundamental issues are the following:

- Further experimental evidence is required to validate the predictions.
- As all reviewers indicate, the performance of the method should be evaluated using separate training and validation datasets. This is an important issue and should be carefully addressed.
- Statistical support needs to be provided for the presented findings.

Moreover the reviewers raise some technical issues and refer to the need to provide several clarifications and/or further details throughout the manuscript.

If you feel you can satisfactorily deal with these points and those listed by the referees, you may wish to submit a revised version of your manuscript. Please attach a covering letter giving details of the way in which you have handled each of the points raised by the referees. A revised manuscript will be once again subject to review and you probably understand that we can give you no guarantee at this stage that the eventual outcome will be favorable.

REFeree REVIEWS

Reviewer #1:

In this work, Arriet-Ortiz et al. are integrating transcriptomics data from two *B. subtilis* datasets and known regulatory interactions to build an expanded transcriptional regulatory network for the organism. They then use KO expression data to computationally validate their findings (1485 out of 2030 tested were "confirmed"). The manuscript is generally well-written, the topic is interesting and the work comes from groups with seminal experimental and computational contributions in the field. I would very much like to see this work published after the points raised here have been addressed.

The Good

- This work exemplifies how computational modeling and experimentation can be integrated with a result that is more than the sum of both parts. I liked how the authors first collected a dataset of 354 profiles on 27 conditions, merged it with an existing dataset of about the same size, used the result compendium with a probabilistic model to predict new interactions, predicted the TF motifs and performed several experiments to inform on the validity of the predictions.

- The authors provide a (rather short but still informative) analysis of the information we get from the merged dataset, regarding TF-target gene correlation and sigma factor binding.

- The computational method they used is state-of-the-art and is based on previously published inferelator and BBSR algorithms. The authors make a convincing case that the algorithm is robust to noise in the prior and it has an improved performance over previous methods.

- The experiments performed with FM and TEM provide experimental evidence on the effect of TF KO on the bacterial phenotype and organization.

The Bad

- The computational inference method is based on calculating TFA (TF activities). Although it is not clear while reading the main manuscript on what these activities are and what they correspond to, their calculation is clear in the methods and supplementary material, where the authors estimate them through linear regression of the expression profiles and the known regulatory interactions (P matrix) from Subtiwiki. Then they go on and infer new relationships by using a heuristic that selects the top regression models from an informed subset of all regression models possible. A potential major flaw, lies with the validation. Is the whole P matrix being used to find the TFA and create the priors? Or are the authors holding off some of the interactions at that stage? The manuscript implies that the whole P is used, which will be an indirect way to use the ground truth for the test set, which is obviously wrong. I emphasize the word "potential", it might be the case that this point is just not explained well. If this is the case, please explain with what data exactly the model has been trained and tested at each step. Otherwise the validation has to be performed again. At any case, please revise the appropriate sections.

- The authors should use K-fold cross-validation (LOO is preferred for model selection, K= 5 or 10 is preferred for validation, see Breiman and Spector, 1992; Kohavi, 1995; Arlot and Celisee, 2010) and measure the bias and variance of their prediction. In addition, whenever bootstrapping is used, please increase the number of subsets, as 50 is not close to be enough (1000 or more if possible). These metrics (and other like std.err., CI etc.) are paramount for making sure that the model is stable given the data, there is minimal overfitting, bias/variance. It is a nuisance to put in place methods for parameter selection, but such methods are paramount for ensuring the validity of the results and it is our responsibility to provide an example to follow.

- The authors compare with GENIE3, CLR and ARACNE. While GENIE3 is a top performing method, CLR and ARACNE are obsolete methods and are known to under-perform (especially ARACNE). Instead the authors should use the Marbach et al. 2012 consensus method or build a custom consensus method from the top few performing methods so far (including for example TIGRESS, SIRENE, etc.). This is the current state-of-the-art and it would be interesting to see how their method compares to each of these methods individually and as a consensus for this dataset.

- The "validation" is only computational with KO expression. The authors claim in their abstract that they "tested 2030 predictions and confirmed 1485 regulatory interactions". From the SOM, it states that a target is validated if "it was differentially expressed with an absolute log-fold change >1 ". This is definitely not adequate and there is no statistical support for these predictions. For an interaction to be "confirmed", it is usually the norm (e.g. see RegulonDB) that two strong experimental evidence sources exist, so this statement is misleading. For a publication in MSB, the authors should perform experimental validation with the standard experimental methods (e.g. Y1H, ChIP-Seq, etc.) of their predictions. I doubt that such a high number of predictions are actually correct. Ideally, the authors can perform validation for all 389 "confirmed" new interactions, but instead focusing on the top 100 predictions is adequate to add both knowledge for the community and a statistical measure of what is to be expected from the method.

- The manuscript is utterly lacking statistics. Please provide p-values by performing the appropriate statistical tests.

The Ugly

This is generally a nicely written manuscript, so most of these points are minor.

- The abstract should be re-written after the validation mentioned above is performed. In addition, it is not clear if any methods are "new" vs. a notable engineering approach to make existing methods work on these datasets (if otherwise, clearly state what has never been introduced before).

- Please report the number of samples (269) and other info for the second dataset in line 127. By reading that paragraph, the reader should have a complete picture of the data available and the dimensionality of the problem.

- TF activation (TFA) is a central point for performing the predictions and it is not clear by reading the main manuscript what this quantity is and what it corresponds to. Please have it better defined and explained, so that the striking difference in correlations to target transcription shown at Fig. 2B can be understood.

- In the "Motivation for estimating TFAs", I found the explanation for moderate correlation (delay) sound but speculative. The authors can mine the data and see if the delay-based hypothesis stands in the case of FFL for *B. subtilis*

- Figure 1, legend for navy blue bars is missing

- Figure 2, in the case of ComK, we see that there is a linear relationship between activity and target transcription. However, ComK target promoters contain two ComK-dimer binding sites, so non-linearity can be expected, so the authors should justify how linearity emerges when transcription levels is replaced by activity.

- In several parts of the manuscript, it states that with BBSR the authors check all possible models - for example in SOM, "we compute all regression models for a given gene corresponding to the inclusion and exclusion of each potential predictor". This is inaccurate since the authors use a heuristic method to sample only a subset of possible regression methods, please revise.

- The manuscript is well referenced, but some of the latest references are missing, e.g. the latest publications from the Baliga, Palsson (O'Brien, Feist), Tagkopoulos, Price labs.

Reviewer #2:

This is a well-written manuscript describing a novel methodology for incorporating known transcription factor (TF) targets into transcriptional regulatory network (TRN) inference by using the responses of the known targets to infer TF activities (TFAs). The new method represents a nicely-motivated way for incorporating known TRN interactions (particularly for large regulons) into the GRN inference procedure, and significantly improves the recovery of regulatory edges that were set aside in training. Additional performance measures were computed using a compendium of TF knockout (TFKO) expression profiles, which were also included in the training of the model.

Concerns:

1. The manuscript includes a good introduction to methods for TRN inference, but does not summarize the literature on estimation of TFAs, for which a fair amount of previous work exists.

2. There is an ambiguity (or at least not clearly spelled out in the paper) which data were used in training and which were excluded and/or subsequently used for model assessment. A particular concern is the inclusion of the TFKO data in the model inference, while it is used subsequently for

performance evaluation. While this would not be a major concern for "standard" TRN model inference, the fact that these data are incorporated in the TFA estimation raises a red flag. The authors state that they compared inferred models which used the TFKO data and did not, and there was 88% agreement (for sporulation sigma factors), which begs the question of why they did not just use the TFKO-excluded data for training of the model (at least the version which was compared against the TFKO data).

3. The estimation of the TFA for a given TF seems to me to be basically a model summarizing the expression profiles of the TF's target genes. Thus, while it definitely makes a lot of sense to do so, using "TFAs" rather than TF expression profiles is another way of stating that you are using the expression profiles of some of a TF's targets to predict additional targets (with similar profiles). This is along the lines of the goals of clustering and other previously-described methods (including the DISTILLER method, mentioned by the authors).

4. The authors primarily used the largest regulons for both (1) estimation of recall, and (2) experimental validation via TFKO data. Why? What was the trend in terms of performance on smaller regulons (shown in Table 1 but not summarized)?

5. The authors summarize the recall/precision of the largest regulons in their predicted network versus the TFKO data (0.73 accuracy). For comparison, what is the accuracy of the input (measured) network versus the TFKO data?

6. Evaluation: There is no assessment of the so-called TFAs, or confirmation that they actually measure activities (e.g. via condition-specific ChIP or similar methods).

7. It would be nice (in the conclusion?) if the authors could present guidelines as to how complete a known TRN is required to obtain similar results in other organisms, as the *B. subtilis* known TRN is essentially unrivalled (in terms of completeness) compared to other model organisms.

8. It would be helpful to see a generalization of the evaluations of this methodology to another species (e.g. one of the "gold standard" data sets from Marbach et al., 2012).

Reviewer #3:

The manuscript by Arrieta-Ortiz and colleagues 'A validated global transcriptional regulatory network ...' deals with the extension of an existing transcriptional regulatory network (TRN) for *B. subtilis* by combination and application of (primarily existing) computational methods to a large compendium of dynamic transcriptomics data in order to infer a large set of previously unknown transcriptional regulatory interactions in this organism. The analysis is based on a previously published and a newly measured compendium of gene expression data under a variety of conditions. The computational analyses incorporates the estimation of transcription factor (TF) activities (termed network component analysis in the literature) and incorporation of this prior knowledge (via the gold standard network used for inferring TF activities) increases accuracy of network inference in a cross-validation scenario substantially. Network validation, including follow-up experiments for detailed analysis leads to the conclusion that the reconstructed network enables new biological insights even for well-characterized regulons, for which the experimental analysis of sporulation (spore polysaccharide synthesis) provides an example.

The study's main contributions are the substantially extended TRN for *B. subtilis* as a well-studied model organism, the validation of a large fraction of newly predicted regulatory interactions, and the detailed analysis of sporulation genes demonstrating the utility of the TRN for subsequent focused experimental studies. The computational aspects appear less novel (i.e., a combination of existing approaches; abstract and discussion would need to be tuned down in this regard) but well-motivated and, in particular, it suggests the incorporation of TF activity estimation as an approach to increase the accuracy of TRN inference. The comments below address primarily aspects of validation and presentation:

Major comments:

(i) Network validation (l.193ff): A recall of 77% of the gold standard (GS) network is certainly high, but an analysis of the causes for missing recall for the remainder of the GS network, despite it being used as prior information, e.g. for TF activity inference, is missing. Similar to novel interactions, an analysis of interactions not recovered from the GS network (e.g., functional enrichments) should be conducted to, for example, assess if the transcriptomics data set was simply not informative with respect to these interactions. It could also be possible to test robustness of newly predicted interactions via setting corresponding GS network interactions in the inferred TRN as fixed. In addition, the authors argue that the use of KO data in training and validation set would be admissible because 'they represent less than 5% of the data ..' (l.230). Since the amount of data is not necessarily related to its information content with respect to inference, this statement is not sensible. Instead, training and validation data set should be clearly separated, and the corresponding key numbers for precision should be reported.

(ii) Comparison of methods performance: This crucial aspect, given the plethora of published TRN inference methods, is currently represented only in Fig. S4 and it appears that at least a discussion of the relative performance (or figure) should appear in the main text. Currently missing aspects include details on numbers of novel interactions and recovery of GS network interactions as well as comments on prediction accuracies in KO tests. A comment on the selection of methods for comparison (e.g., Fig. S4B) should also address if these methods are tailored to the use of dynamic data (original steady-state versions of the inference methods have been cited).

(iii) Robustness analysis (l.183ff): The robustness analysis consists of using 50% GS network interactions (~1.400) and additional 100 ... 500 false interactions, where there is a clear drop in performance (Fig. 3B) already at the lowest level of contamination. In addition to a misleading labeling of the x-axis in this figure, the presented data hardly supports the claims of a 'very high error-tolerance'. The corresponding text should be revised and include a more clear description of the test for robustness (e.g. relating to specific meaning of prior weight, Fig. 3B).

(iv) Reproducibility of the analysis: In particular with respect to the details of incorporating prior knowledge, the description of the method is not sufficient to reproduce the analysis. For example, Extended View, 'Bayesian regression ...' states that values of \bar{g} were set according to additional knowledge, but neither the additional knowledge nor the numerical values used are specified. All pertinent details should be included, ideally in the description and executable code for the entire inference method.

Minor comments:

(i) l.107: The sentence 'By applying a unified new computational method ...' appears overstated w.r.t. novelty of the method and the detailed experimental design is never explained in more depth (a corresponding addition in the first results section would be meaningful).

(ii) l.188: The 'complementary nature of the two data compendia' is not demonstrated; please elaborate / provide evidence of re-phrase the sentence.

(iii) l. 248ff.: Please comment on cluster characteristics for the remaining 3 clusters without functional enrichment.

(iv) l.405 ff.: The presentation of the inference method to a large extent overlaps between Materials and Methods and Extended Text - I suggest to combine text in only one of the two places. Please also clarify the following: (a) Extended Text, GS network: How were the specific regulons (especially those relevant to the detailed experimental analysis) manually curated? (b) The first two equations in 'Bayesian regression ..' are incomplete.

Reviewer #1:

- The computational inference method is based on calculating TFA (TF activities). Although it is not clear while reading the main manuscript on what these activities are and what they correspond to,

their calculation is clear in the methods and supplementary material, where the authors estimate them through linear regression of the expression profiles and the known regulatory interactions (P matrix) from Subtiwiki. Then they go on and infer new relationships by using a heuristic that selects the top regression models from an informed subset of all regression models possible. A potential major flaw, lies with the validation. **Is the whole P matrix being used to find the TFA and create the priors? Or are the authors holding off some of the interactions at that stage?** The manuscript implies that the whole P is used, which will be an indirect way to use the ground truth for the test set, which is obviously wrong. I emphasize the word "potential", it might be the case that this point is just not explained well. If this is the case, please explain with what data exactly the model has been trained and tested at each step. Otherwise the validation has to be performed again. At any case, please revise the appropriate sections.

This point was raised by two of the reviewers and is, we agree, a major flaw in the description of the methods in the original manuscript. The flaw was in the manuscript and not the method. References to TFA's estimation and Network Component Analysis were added to the main text. Additionally, a paragraph summarizing the main advances in TFA estimation and applications was added in the introduction section (p.5, l.133-150).

When evaluating the performance increase obtained by using TFA (Fig. 3A) and the robustness to noise in the prior (Fig. 3B), we do not use the entire gold standard as prior. Instead, we randomly choose 50% of the gold standard to use as prior and use the remaining 50% as the test set. Precision and recall are then calculated excluding any interactions that were also in the prior, i.e. only "novel" interactions are evaluated. However, for the final network, as well as the networks that exclude specific KO-related arrays, all gold standard interactions are used as prior to allow for the most accurate inferred networks. When we use the KO data to evaluate the networks, we specifically show the performance for novel interactions which are not present in the gold standard (new Fig. 3C, right, and new Fig.4).

We have updated the relevant sections in the main text, figures, and supplement to clarify these scenarios.

- **The authors should use K-fold cross-validation** (LOO is preferred for model selection, K= 5 or 10 is preferred for validation, see Breiman and Spector,1992; Kohavi, 1995; Arlot and Celisee, 2010) and measure the bias and variance of their prediction. **In addition, whenever bootstrapping is used, please increase the number of subsets, as 50 is not close to be enough (1000 or more if possible).** These metrics (and other like std.err., CI etc.) are paramount for making sure that the model is stable given the data, there is minimal overfitting, bias/variance. It is a nuisance to put in place methods for parameter selection, but such methods are paramount for ensuring the validity of the results and it is our responsibility to provide an example to follow.

As suggested by the reviewer, we did increase the number of bootstraps; however, our analysis showed that there is no significant change after 100 bootstraps (new Fig S3) and, therefore, 1000 bootstraps are not required (p. 8, l. 243-247): "Next, to evaluate if the number of bootstraps affected the output of the inference approach we compared the top 5000 interactions for inferred networks using 2 up to 100 bootstraps in the BSb1 dataset, PY79 dataset or both (combined) to the top 5000 interactions using one less bootstrap (Fig. S3). We observed that, for all networks, after 20 bootstraps more than 4890 (97.8%) are shared when another bootstrap is added. This finding suggests a rapid convergence of the error estimates computed by BBSR-TFA".

Moreover, the confidence score, associated to each interaction, also contains information about the frequency of an interaction in the bootstraps (p.19, l.558-559): "The final confidence score is defined as the mean position of the interaction (among all bootstraps) when the interactions are ranked by the variance explained".

- The authors compare with GENIE3, CLR and ARACNE. While GENIE3 is a top performing method, CLR and ARACNE are obsolete methods and are known to under-perform (especially ARACNE). **Instead the authors should use the Marbach et al. 2012 consensus method or build a custom consensus method** from the top few performing methods so far (including for example TIGRESS, SIRENE, etc.). This is the current state-of-the-art and it would be interesting to see how their method compares to each of these methods individually and as a consensus for this dataset.

To address this issue, we have included a consensus method (called META) which is the rank combined prediction of CLR, GENIE3 and our approach BBSR (new Fig. 3C). We also used SIRENE but could only produce results that were close to random. In fact, even for the example *E. coli* dataset (Marbach *et al.* 2012) and script provided with the SIRENE archive, results seemed to be random. We have contacted the authors but in one case received no response, and in the other case the author could not run SIRENE himself due to missing prerequisites. Therefore, we decided not to include SIRENE.

In addition to a new Fig. 3C, a section comparing the performances of BBSR-TFA and the other used methods was added to the main text (p. 9, l. 277-285): “We performed the same evaluation of the GENIE3 and CLR networks, as well as a consensus method (META) that rank combines the prediction of the three methods (as suggested by Marbach *et al.*, 2012). The performance of all methods is shown in Fig. 3C (analysis by regulon is shown in Fig. 4 and Fig. S4). We observed that BBSR-TFA outperforms the other methods with respect to the fraction of supported predictions, when all predictions (left panel) or only novel interactions (right panel) are considered, and all methods greatly benefited from using TF activities. The *Inferelator* (both BBSR and BBSR-TFA version) is the most conservative method in predicting novel interactions, which is a result of including prior information in the model selection step. This also resulted in the lowest absolute and proportional number of unsupported novel predictions.”

- The "validation" is only computational with KO expression. The authors claim in their abstract that they "tested 2030 predictions and confirmed 1485 regulatory interactions". **From the SOM, it states that a target is validated if "it was differentially expressed with an absolute log-fold change >1". This is definitely not adequate and there is no statistical support for these predictions.** For an interaction to be "confirmed", it is usually the norm (e.g. see RegulonDB) that two strong experimental evidence sources exist, so this statement is misleading. For a publication in MSB, the authors should perform experimental validation with the standard experimental methods (e.g. Y1H, ChIP-Seq, etc.) of their predictions. I doubt that such a high number of predictions are actually correct. Ideally, the authors can perform validation for all 389 "confirmed" new interactions, but instead focusing on the top 100 predictions is adequate to add both knowledge for the community and a statistical measure of what is to be expected from the method.

*We thank the reviewer for raising this issue. This is an important point that we did not discuss in sufficient detail in the original version of the manuscript. Because we relied primarily on SubtiWiki (which did not edict standard guidelines about what constitutes a confirmed regulatory interaction), we included in the gold standard many putative interactions that are not necessarily backed up by two strong experimental sources. As a consequence, we also did not follow a strict protocol to call a prediction confirmed in the original version of the manuscript. We are in full agreement with the reviewer that the term “confirmed” is too strong and as a result, we thoroughly edited the text to replace the term “confirmed” by “experimentally supported”. We also changed the title of the paper from “A Validated Global Transcriptional Regulatory Network for *Bacillus subtilis*” to “An Experimentally Supported Model of the *Bacillus subtilis* Global Transcriptional Regulatory Network”.*

*In addition, we focused our follow-up analyses on the top 500 novel predictions (new file Dataset EV5, see below) by gathering from the published literature every possible type of information (ChIP-seq, transcriptomics, functional characterization, operon structure, presence of putative binding sites in the promoter) that would support our predictions and our transcriptional analyses using KO strains. Additional experiments would require a significant time investment (probably a whole year) and, at this stage, we do not have the necessary financial resources for such a large scale ChIP-seq validation effort. We point out, however, that in the past couple of years ChIP seq data have been published for about 10 *B. subtilis* TFs (*AbrB*, *CcpA*, *CodY*, *PhoP*, *Spx*, *ResD*, *TnrA*, *WalR* and *Zur*) and, in the updated Table I and new files Datasets EV4 and EV5, we incorporate this information in the analysis of our predictions.*

We also understand the reviewer's concern about the statistical significance of a 2 fold-change ratio in transcriptional profiling for defining differentially transcribed genes (see also the next point about statistics). Our experimental design focused on maximizing the number of different conditions that we could query at the expense of more replicates. So, in many cases, we only performed the KO

analyses in duplicate, which limits our options for the statistical analysis of the data. Nevertheless, we collected additional KO data replicates for SigD, SigG and SigW (and we already had data in triplicate for SigB and SigM). As a result, the p-values associated with differential expression tests are often not low enough to sustain correction for multiple testing.

As explained in the supplement (section on differential gene transcription analysis): “We compared gene expression profiles of wild-type (WT) and the respective mutant strains (KO) using Bayesian t-tests with the Cyber-T tool (Baldi and Long 2001). The main challenge we faced with the Bayesian t-tests was the low number of differentially expressed genes detected for several TFs after correcting for multiple hypothesis testing. This issue was also observed when other tools such as Limma (Smyth 2004) were used. To evaluate if the small number of detected differentially expressed genes was due to the lack of change in the expression profile of expected genes (based on the regulon's information derived from the GS network), we computed the Precision Recall plot for each WT and KO comparison. We ranked the genes using the raw p-values obtained from the corresponding Bayesian t-test (in increasing order) and computed the Area Under Precision Recall (AUPR) curve using the TF targets in the GS network as reference. To get the expected distribution of AUPRs for the case where the arrays were uninformative with respect to the expected set of differentially expressed genes, we also computed the AUPR using multiple random rankings of genes.”

For time series data, all TFs but SigH, SigL and Spo0A have at least one time point with AUPR scores significantly higher than the ones observed in the null distribution. These findings suggest that information derived from the raw p-values is still meaningful and agree with the expectations derived from GS network. Therefore, we used a raw p-value threshold of 0.01 to consider a gene differentially expressed and supported by the KO data (p.20, l.578-582): “We compared gene expression profiles of wild-type (WT) and the respective mutant strains (KO) using Bayesian t-tests with the Cyber-T tool (Baldi & Long, 2001). We followed the authors' recommendation to keep the number of replicates plus the confidence parameter equal to ten. Genes with p-values equal or smaller than 0.01 were considered differentially transcribed”.

*As indicated above, we analyzed the top 500 novel interactions (ranked by the associated confidence score). We added the following section to the main text (pp.12-13, l.346-364):” We also analyzed the top 500 novel predicted interactions in the final combined network (ranked by the associated confidence score, **Dataset EV5**). The top 500 novel predictions include 483 target genes and 91 TFs. Forty-one of these interactions have been validated by external sources since compilation of the GS and five have been validated in the current study using GFP fusions (see below). Seventy-six of the remaining interactions can also be validated on the grounds that the genes involved belong to operons that include previously known targets. This applies to many short genes that were added to genome after re-annotation of the *B. subtilis* 168 sequence (Barbe *et al*, 2009). These genes were absent from microarrays generated from the original genome annotation (Kunst *et al*, 1997) and would have been missed in transcriptional profiling experiments conducted prior to the year 2010. Considering the remaining 378 interactions, transcriptional profiling data was available (from this and previous studies) for 210 interactions. We found that 153 out of these 210 interactions (i.e. 73%) were experimentally supported by transcriptional profiling data (p-value <0.01 and/or external validation). In parallel, we performed a search and/or collected information from previous studies for the presence of putative binding sites for TFs involved in 193 putative interactions. A sequence motif compatible with a previously reported consensus binding site was identified in the corresponding promoter sequences for 136 out of these 193 interactions (71%), thus providing additional evidence for these predictions. In total, there were 144 interactions for which both KO and motif data were available. Out of these 144 putative interactions, 120 (83%) were supported by both (in addition to the 122 predictions that we considered already validated by external sources). These findings suggest high prediction accuracy for the top 500 predictions (when ranked by confidence score).”*

- Please provide p-values by performing the appropriate statistical tests.

See also above. P-values associated to the statistical tests for differential gene expression analyzes are included in the updated Dataset EV4. The e-values associated to all reported motifs are shown in the new Fig S5. We also included confidence scores for each prediction. This score can be used for ranking the interactions.

This is generally a nicely written manuscript, so most of these points are minor.

- The abstract should be re-written after the validation mentioned above is performed. In addition, it is not clear if any methods are "new" vs. a notable engineering approach to make existing methods work on these datasets (if otherwise, clearly state what has never been introduced before).

We re-wrote the abstract as suggested (p.2, l.55-61): "Taking advantage of the large number of known regulatory interactions in *Bacillus subtilis* and two transcriptomics datasets (including one with 38 separate experiments collected specifically for this study), we use a new combination of network component analysis and model selection to simultaneously estimate transcription factor activities and learn a substantially expanded transcriptional regulatory network for this bacterium. In total, we predict 2258 novel regulatory interactions and recall 74% of the previously known interactions. We obtained experimental support for 391 (out of 635 evaluated) novel regulatory edges (62% accuracy), thus significantly increasing our understanding of various cell processes, such as spore formation".

We also added a section to the introduction highlighting the previous work related to TF activity estimation (p.5, l.133-150).

In addition, we now clearly state that our method is a new combination of existing approaches (p. 6, l. 172-173): "we used a new combination of our *Inferelator-BBSR* approach (Greenfield *et al*, 2013), with a method for estimating transcription factor activities (TFAs)".

- Please report the number of samples (269) and other info for the second dataset in line 127. By reading that paragraph, the reader should have a complete picture of the data available and the dimensionality of the problem.

Done as suggested by the reviewer (p.6, l.165-167): "We added a previously published dataset with 269 samples covering 104 conditions, using strain BSB1, another derivative of strain 168 (Nicolas *et al*, 2012)".

- TF activation (TFA) is a central point for performing the predictions and it is not clear by reading the main manuscript what this quantity is and what it corresponds to. Please have it better defined and explained, so that the striking difference in correlations to target transcription shown at Fig. 2B can be understood.

We modified the introduction as follows (p.5, l.139-141): "These methods have in common that they model gene expression to be the result of the connectivity strength between TF-gene pairs and TF activity, where the activity is a latent variable pooling the effects of post-transcriptional and post-translational modifications."

Additionally, the following section was added to the description of Fig 2B (p.8, l.206-211): "By contrast, for both ComK and CodY, a linear relationship is observed between TF activity and transcription of its targets. This linear relationship is a consequence of the way TFAs are estimated (see methods). Considering that the *Inferelator* is based on a linear model (see methods), this linearization step is likely to improve the detection of additional regulatory interactions. This improvement would affect primarily TFs whose activity can be accurately estimated (i.e. those with at least 10 known target genes, see below, **Fig. S2**)."

- In the "Motivation for estimating TFAs", I found the explanation for moderate correlation (delay) sound but speculative. The authors can mine the data and see if the delay-based hypothesis stands in the case of FFL for *B. subtilis*

The delay-based hypothesis in incoherent FFLs is supported, theoretically and experimentally, by many publications (see Mangan and Alon 2003, Alon 2007). We added these references to the text (p.7, l.196).

- Figure 1, legend for navy blue bars is missing

The navy blue bars were deleted from the figure.

- Figure 2, in the case of ComK, we see that there is a linear relationship between activity and target transcription. However, ComK target promoters contain two ComK-dimer binding sites, so non-linearity can be expected, so the authors should justify how linearity emerges when transcription levels is replaced by activity.

The linearity is observed in the comparison between ComK estimated activity and the transcription level of its targets. This is a consequence of the approach used for estimating TF activities (as explained in the methods section). We are not claiming that the relationship between TFs and their targets should be linear (in fact the TFA method is needed in part for this reason). It is important to emphasize though that the linearization performed by the TFA step may increase the performance of the Inferelator (or of any other inference method that assumes a linear relationship between predictor values and target gene expression). A clarification has been added at the end of the paragraph describing Fig 2B (p.7, l.208-211): "Considering that the Inferelator is based on a linear model (see methods), this linearization step is likely to improve the detection of additional regulatory interactions. This improvement would affect primarily TFs whose activity can be accurately estimated (i.e. those with > 10 known target genes, see below, Fig. S2)."

- In several parts of the manuscript, it states that with BBSR the authors check all possible models - for example in SOM, "we compute all regression models for a given gene corresponding to the inclusion and exclusion of each potential predictor". This is inaccurate since the authors use a heuristic method to sample only a subset of possible regression methods, please revise.

The reviewer is correct. In the supplement, the sentence has been changed to: "We now describe the BBSR method, an inference method that computes the regression models for a given gene i corresponding to the inclusion and exclusion of each TF that has a known regulatory effect on i, and the ten TFs with highest time-lagged CLR"

- The manuscript is well referenced, but some of the latest references are missing, e.g. the latest publications from the Baliga, Palsson (O'Brien, Feist), Tagkopoulos, Price labs.

These references have been added to the introduction section (p.4, l.107-121).

Reviewer #2:

Concerns:

1. The manuscript includes a good introduction to methods for TRN inference, but does not **summarize the literature on estimation of TFAs**, for which a fair amount of previous work exists.

A similar point was made by reviewer#1 and, as mentioned above, we have added a paragraph about previous work in the field of Network Component Analysis and TFA estimation (p.5, l.133-150).

2. There is an ambiguity (or at least not clearly spelled out in the paper) which data were used in training and which were excluded and/or subsequently used for model assessment. **A particular concern is the inclusion of the TFKO data in the model inference, while it is used subsequently for performance evaluation.** While this would not be a major concern for "standard" TRN model inference, the fact that these data are incorporated in the TFA estimation raises a red flag. The authors state that they compared inferred models which used the TFKO data and did not, and there was 88% agreement (for sporulation sigma factors), which begs the question of why they did not just use the TFKO-excluded data for training of the model (at least the version which was compared against the TFKO data).

A similar point was raised by the other reviewers. To address this issue, we changed our performance evaluation approach as described on p.9 (l.265-275): "To have a clear separation between training and evaluation datasets, we predicted a network for each analyzed regulon using training data that excluded data relevant for the WT and KO strains comparison for that TF. In total

we obtained 17 networks (**Fig. 4**), one for each TF with KO data (we refer to these networks as evaluation networks). The reason for not excluding all KO data at once is that it would represent a 25% decrease in the PY79 dataset size. For each TF and the corresponding set of KO conditions, we tested all genes for differential transcription (DT) using Bayesian t-tests. We considered all genes with p-values < 0.01 as DT (see methods for details). Genes that were predicted as targets in the TF-specific evaluation network were considered true positives (i.e. supported by the KO data) if they were DT, while targets that were not DT were considered false positives (i.e. not supported by the KO data).”

Performance of BBSR and other inference methods is shown in new Fig. 3C.

3. The estimation of the TFA for a given TF seems to me to be basically a model summarizing the expression profiles of the TF's target genes. Thus, while it definitely makes a lot of sense to do so, using "TFAs" rather than TF expression profiles is another way of stating that you are using the expression profiles of some of a TF's targets to predict additional targets (with similar profiles). This is along the lines of the goals of clustering and other previously-described methods (including the DISTILLER method, mentioned by the authors).

The paper is aimed at providing an improved and more comprehensive regulatory network model for a key model organism. Thus, we did not aim to present an entirely novel method. That said, the specific method presented (TFA plus our approach to model selection and use of time series) is new. A key difference between DISTILLER and our own previous efforts with the Baliga lab (cMonkey+Inferelator) is that the estimated activities can then be used to find new targets for individual genes or regulons. There are several key differences (single gene models, activity estimates that represent single and not group activities, and the ability to constrain model building with prior information in a data-driven manner). The current methods formulation also outperforms prior methods by a substantial margin. This performance gain is also substantiated by prior works that include double blind tests of our methods and tests of our methods by third parties.

4. The authors primarily used the largest regulons for both (1) estimation of recall, and (2) experimental validation via TFKO data. **Why? What was the trend in terms of performance on smaller regulons (shown in Table 1 but not summarized)?**

We chose to validate the larger regulons first because of biological and efficiency concerns. We do detail how performance changes with regulon size, showing that TFA improves for regulons with greater than 10 targets (p.10, l.293-297): “Since transcriptional profiles instead of TFAs were relied upon for TFs with no previously known targets, the average number of novel predictions was low for this group (4.7 per TF for TFs with no priors, compared to 22.5 per TF for TFs with more than ten priors). Similarly, ranking positions of novel interactions for TFs with less than six known targets were much lower than those for TFs with more than ten targets (Table S3)”.

See also (p.8, l. 239-241): “The vast majority of TFAs are stable (as indicated by the distributions of the pair-wise correlations of the activities; Fig. S2) and TFs with ten or more priors have more stable estimated TFAs than TFs with less than 10 priors”.

5. The authors summarize the recall/precision of the largest regulons in their predicted network versus the TFKO data (0.73 accuracy). **For comparison, what is the accuracy of the input (measured) network versus the TFKO data?**

The KO support rate for the full and recovered GS network is 0.7 and 0.62, respectively. These values have been added to p.12, l. 343-344.

6. **Evaluation:** There is no assessment of the so-called TFAs, or confirmation that they actually measure activities (e.g. via condition-specific ChIP or similar methods).

We chose to focus on validation of interactions over activities, and consider a specific validation of this step via additional experiments beyond the scope of the current paper. We did explore these activities (comparing estimated activities to conditions for several well characterized TFs, in the main text and supplement) and show reasonable trends (activities are less related to expression for TFs with auxiliary activators and for TFs that respond to highly variable environmental parameters

like cell density). It is important to note, however, that TFAs are indirectly validated as part of the inference pipeline (showing improved out of sample performance compared to raw expression) and directly as part of our inference pipeline.

We have added text outlining TFA direct validation as a key future direction (p.16, l.463-465): “In future network inference attempts, the inclusion of complementary data types should be prioritized, especially genome-wide binding assays (for poorly characterized TFs) and proteomics (to characterize post-transcriptional regulatory events)”.

7. It would be nice (in the conclusion?) if the authors could present guidelines as to how complete a known TRN is required to obtain similar results in other organisms, as the *B. subtilis* known TRN is essentially unrivalled (in terms of completeness) compared to other model organisms.

We agree that this is a key question to begin addressing and the last sentence of the conclusion hints at this (p. 16, l.472-474): “Overall, the strategy delineated here can be applied to other bacteria and eukaryotic cells as long as a minimal set of priors and large transcriptional datasets are available”.

Furthermore, new Fig.S2 (detailing how TFA stability changes with the number of known targets) begins to explore this important question. In general, and as mentioned above, TFs with 10 or more targets provide stable TFA estimates. This would suggest that an optimal experimental design for future works would aim KO and ChIP experiments at the least well studied TFs.

8. It would be helpful to see a generalization of the evaluations of this methodology to another species (e.g. one of the "gold standard" data sets from Marbach et al., 2012).

We have done this in prior papers and are planning to use this code in collaboration with other groups on other bacterial and eukaryotic species, but feel this paper is already bursting at the seams and thus additional organisms are beyond the scope of the current paper.

Reviewer #3:

Major comments:

(i) Network validation (l.193ff): A recall of 77% of the gold standard (GS) network is certainly high, **but an analysis of the causes for missing recall for the remainder of the GS network, despite it being used as prior information, e.g. for TF activity inference, is missing.** Similar to novel interactions, an analysis of interactions not recovered from the GS network (e.g., functional enrichments) should be conducted to, for example, assess if the transcriptomics data set was simply not informative with respect to these interactions. It could also be possible to test robustness of newly predicted interactions via setting corresponding GS network interactions in the inferred TRN as fixed. In addition, the authors argue that the use of KO data in training and validation set would be admissible because 'they represent less than 5% of the data ..' (l.230). Since the amount of data is not necessarily related to its information content with respect to inference, this statement is not sensible. **Instead, training and validation data set should be clearly separated, and the corresponding key numbers for precision should be reported.**

This comment has been partially addressed above. The support rate of KO data for the interactions present in the GS and not included in our predicted model was evaluated as suggested by the reviewer and added as follows (p.12, l.338-344): “Because not every interaction in the GS was recalled in our model, we checked whether the target genes in these interactions were differentially transcribed in the corresponding KO experiment. The global rate of differential transcription for these missing interactions (i.e. those present in the GS network but absent in our model) was 0.38, suggesting that at least some of the interactions in the GS may either be inaccurate or strictly dependent on strain background and specific experimental conditions. In any event, this number is significantly lower than the support rate for the full predicted network (0.7), the set of interactions recovered from the GS (0.74) or the set of novel interactions (0.62).”

See also above response to the other reviewers; p.9 (l.265-275): “To have a clear separation between training and evaluation datasets, we predicted a network for each analyzed regulon using training data that excluded data relevant for the WT and KO strains comparison for that TF. In total

we obtained 17 networks (**Fig. 4**), one for each TF with KO data (we refer to these networks as evaluation networks). The reason for not excluding all KO data at once is that it would represent a 25% decrease in the PY79 dataset size. For each TF and the corresponding set of KO conditions, we tested all genes for differential transcription (DT) using Bayesian t-tests. We considered all genes with p-values < 0.01 as DT (see methods for details). Genes that were predicted as targets in the TF-specific evaluation network were considered true positives (i.e. supported by the KO data) if they were DT, while targets that were not DT were considered false positives (i.e. not supported by the KO data).”

(ii) Comparison of methods performance: This crucial aspect, given the plethora of published TRN inference methods, is currently represented only in Fig. S4 and it appears that at least a discussion of the relative performance (or figure) should appear in the main text. Currently missing aspects include details on numbers of novel interactions and recovery of GS network interactions as well as comments on prediction accuracies in KO tests. A comment on the selection of methods for comparison (e.g., Fig. S4B) should also address if these methods are tailored to the use of dynamic data (original steady-state versions of the inference methods have been cited).

As suggested by this reviewer and as explained above, a paragraph comparing TFA-BBSR to other inference methods has been incorporated in the main text (p. 9, l.277-285), we modified the corresponding figure (Fig 3A and Fig.3C) and added a new Fig.4. We have included a consensus method (called META) which is the rank combined prediction of CLR, GENIE3 and our approach BBSR. In summary, BBSR-TFA outperforms the other methods with respect to the fraction of supported predictions, and all methods greatly benefited from using TF activities. The support rate for the full set of tested predictions and novel predictions is 0.61 and 0.44 (Fig.3C), respectively. The Inferelator (both BBSR and BBSR-TFA version) is the most conservative method in predicting novel interactions, which is a result of including prior information in the model selection step. This also resulted in the lowest absolute and proportional number of unsupported novel predictions. All tested methods used the exact same design and response matrices. The use of dynamic data was as described in the method section of our approach (p.19, l. 532-537). We expanded the method section to better explain this (other inference methods, p.20, l. 568-570).

(iii) Robustness analysis (l.183ff): The robustness analysis consists of using 50% GS network interactions (~1.400) and additional 100 ... 500 false interactions, where there is a clear drop in performance (Fig. 3B) already at the lowest level of contamination. In addition to a misleading labeling of the x-axis in this figure, the presented data hardly supports the claims of a 'very high error-tolerance'. **The corresponding text should be revised and include a more clear description of the test for robustness** (e.g. relating to specific meaning of prior weight, Fig. 3B).

*The corresponding text has been changed to (p.9, l.249-254): “To assess robustness to noise (i.e. presence of incorrect or irrelevant edges in the GS), we used as priors 50% of the GS interactions (randomly selected) and added various amounts of random false interactions (**Fig. 3B**). Performance on the remaining 50% of GS interactions demonstrates a very high error-tolerance and relative insensitivity to input parameters. Specifically, AUPR in the presence of a noisy prior is higher than the no-prior baseline even at a true:false prior ratio of 1:10, if weight in the model selection step (g-prior in the methods section) is not too large (i.e. < 2).”*

The original x-axis labels referred to ratios, i.e. 50:500 meant “50% of gold standard interactions plus 10 times as many random false interactions). The figure legend has been updated to clarify this relationship (p.22, l.641-644):” Performance of network inference (AUPR: area under precision recall curve) on the combined, BSB1 and PY79 networks in the presence of false prior information. 50% of the edges in the GS are used as true priors and various amounts of random edges are added. Performance is evaluated on the leave-out set of interactions. Each point represents the median of 5 random samples of 50% of the GS set”.

(iv) Reproducibility of the analysis: In particular with respect to the details of incorporating prior knowledge, the description of the method is not sufficient to reproduce the analysis. For example, Extended View, 'Bayesian regression ...' states that values of \bar{g} were set according to additional knowledge, but neither the additional knowledge nor the numerical values used are specified. All pertinent details should be included, ideally in the description and executable code for the entire inference method.

The \bar{g} value used for creating the networks described in the text is 1.1. This information has been added to the methods section (p.19, l.545-547): “Prior knowledge is incorporated by using a modification of Zellner’s g-prior (Zellner, 1983) to include subjective information on the regression

parameters. A g-prior equal to 1.1 was used for the combined network described in this study.”

Minor comments:

(i) l.107: The sentence 'By applying a unified new computational method ...' appears overstated w.r.t. novelty of the method and the detailed experimental design is never explained in more depth (a corresponding addition in the first results section would be meaningful).

*The term 'unified' refers to the fact that the method is new in terms of our combined approach to model selection with estimated activities as predictors. We rephrased this sentence as follows (p.5, .148-150): "In this work we apply a unified new combination of NCA and model selection to an experimental design expressly conceived to dynamically probe the principal cellular pathways of *B. subtilis*, and we identify 2260 novel regulatory interactions of unprecedented accuracy."*

(ii) l.188: The 'complementary nature of the two data compendia' is not demonstrated; please elaborate / provide evidence of re-phrase the sentence.

*The sentence has been re-phrased as follows (p.9, l.256-258): "The fact that the combined network has the highest AUPR in **Fig. 3B** indicates that many true interactions that would have been excluded otherwise are recovered from the combination of the PY79 and BSB1 independently predicted networks. This highlights the complementary nature of the two data compendia".*

(iii) l. 248ff.: Please comment on cluster characteristics for the remaining 3 clusters without functional enrichment.

Because the final network model has changed since the previous version of the paper (due to the addition of more experiments and microarrays, the figure has been updated, new Fig. 5) and we have expanded the discussion of each cluster in the figure (p.13-14, l.366-394).

(iv) l.405 ff.: The presentation of the inference method to a large extent overlaps between Materials and Methods and Extended Text - I suggest to combine text in only one of the two places. Please also clarify the following: (a) Extended Text, GS network: How were the specific regulons (especially those relevant to the detailed experimental analysis) manually curated? (b) The first two equations in 'Bayesian regression ..' are incomplete.

The manuscript was edited extensively with these comments in mind. We believe the methods description is both more complete and clearer following the three reviewer suggestions.

2nd Editorial Decision

08 October 2015

Thank you again for submitting your work to Molecular Systems Biology. We have now heard back from the referees who agreed to evaluate your manuscript. As you will see below, the referees are satisfied with the modifications made and they think that the study is suitable for publication.

Before formally accepting the manuscript, we would ask you to address some minor editorial issues listed below.

REFeree REVIEWS

Reviewer #1:

All my comments have been addressed and in a satisfactory manner. I would have wished for new experimental validation but I understand the financial and time constraints. I still think a smaller scale of experimental validation would have been feasible (a couple months and a couple thousand dollars for the Y1H kits). However, the new data curation and analysis the team did adds value and is satisfactory.

Reviewer #2:

No additional or major issues with the updated version of the manuscript.